# Cortex folding by combined progenitor expansion and adhesion-controlled neuronal migration

Seung Hee Chun[1], Da Eun Yoon[2], D. Santiago Diaz Almeida[1], Mihail Ivilinov Todorov[3,4], Tobias Straub[5], Tobias Ruff[6], Wei Shao[7], Jianjun Yang[8], Gönül Seyit-Bremer[1], Yi-Ru Shen[1], Ali Ertürk[3,4,9,10], Daniel del Toro[11], Songhai Shi[8] & Rüdiger Klein[1] ✉

Folding of the mammalian cerebral cortex into sulcal fissures and gyral peaks is the result of complex processes that are incompletely understood. Previously we showed that genetic deletion of Flrt1/3 adhesion molecules causes folding of the smooth mouse cortex into sulci resulting from increased lateral dispersion and faster neuron migration, without progenitor expansion. Here, we show in mice that combining the Flrt1/3 double knockout with an additional genetic deletion that causes progenitor expansion, greatly enhances cortex folding. Expansion of intermediate progenitors by deletion of Cep83 leads to a relative increase in Flrt-mutant neurons resulting in enhanced formation of sulci. Expansion of apical progenitors by deletion of Fgf10 leads to a relative reduction in Flrt-mutant neurons resulting in enhanced formation of gyri. These results together with computational modeling identify key developmental mechanisms, such as adhesive properties, cell densities and migration of cortical neurons, that cooperate to promote cortical gyrification.

One of the most remarkable aspects of brain development is the folding of the cerebral cortex, which is believed to enhance the cognitive capacities of large mammals. Brain size has changed dramatically during mammalian evolution[1] and this is mostly due to an increase in size of the cerebral cortex. The cerebral cortex is highly folded into peaks (gyri) and fissures (sulci) in gyrencephalic species, or has a smooth surface in lissencephalic, typically smaller mammals such as the mouse[2].

Previous studies have identified mechanisms that drive cortex folding, including cell proliferation, migration and biomechanical properties of the tissue[3]. Most studies have focused on amplification of cell proliferation due to the correlation between larger brains and cortical folding[4,5]. Progenitors are distributed in two different cortical germinal layers: the ventricular zone (VZ), which includes the neuroepithelial cells (NECs) and apical radial glia (aRG), and the subventricular zone (SVZ) containing intermediate progenitors (IPs) and

[1]Department of Molecules – Signaling – Development, Max Planck Institute for Biological Intelligence, Martinsried, Germany. [2]Transgenic Core Facility, Max Plank Institute of Biochemistry, Martinsried, Germany. [3]Institute for Stroke and Dementia Research (ISD), University Hospital, Ludwig-Maximilians-University Munich (LMU), Munich, Germany. [4]Institute for Tissue Engineering and Regenerative Medicine (iTERM), Helmholtz Munich, Neuherberg, Germany. [5]Bioinformatics Core, Biomedical Center, Faculty of Medicine, Ludwig-Maximilians University (LMU), Martinsried, Germany. [6]Laboratory of Biosensors and Bioelectronics, Institute for Biomedical Engineering, Eidgenössische Technische Hochschule (ETH) Zürich, Zürich, Switzerland. [7]Biochemistry, Cell and Molecular Biology Allied Graduate Program, Weill Cornell Medical College, New York, NY, USA. [8]New Cornerstone Science Laboratory, IDG/McGovern Institute for Brain Research, School of Life Sciences, Tsinghua University, Beijing, China. [9]Munich Cluster for Systems Neurology (SyNergy), Munich, Germany. [10]Koç Unive Koç University, School of Medicine, İstanbul, Turkeyrsity, School of Medicine, İstanbul, Turkey. [11]Department of Biomedical Sciences, Faculty of Medicine and Health Sciences, Institute of Neurosciences, IDIBAPS, University of Barcelona, Barcelona, Spain. ✉e-mail: ruediger.klein@bi.mpg.de

basal radial glia (bRG)[3]. A massive increase in the surface area of both layers is an evolutionary hallmark of gyrencephalic species. This includes the thickening of the SVZ, which is then subdivided into an inner (ISVZ) and an outer (OSVZ) subventricular zone[6–8]. The SVZ in gyrencephalic species like the ferret is not homogeneous as in the mouse, and displays microdomains with high and low proliferation of progenitors, a pattern that favors the formation of future gyri and sulci, respectively[8]. Supporting this model, local amplification of progenitors in the murine cortex induces the formation of gyrus-like structures[9–12]. Interestingly, studies inducing uniform progenitor expansion in the mouse cortex have observed thickening of this structure without folding[13,14]. These results suggest that progenitor expansion in discrete cortical domains is a key event to induce gyration of the cerebral cortex.

Generating genetic mouse models in which cortex folding is induced by progenitor expansion has remained a challenge. Elevated Sonic hedgehog (Shh) signaling by expressing a constitutively active receptor[15], or increased Wnt signaling through the loss of transcription factors Lmx1a/1b[16], lead to IP expansion and cortex folding in mice. However, these modifications affect the survival of mutant embryos at later developmental and postnatal stages. One exception is the genetic ablation of the centrosomal protein 83 (Cep83). Its deletion removes the apical membrane anchoring of the centrosome in aRGs and induces an expansion of IPs. This progenitor amplification mainly occurs in the anteromedial regions of the cortex, which correlates with cortex folding[17].

Fibroblast growth factor (FGF) signaling is one of the most relevant pathways that regulate cortical proliferation, in addition to Wnt, Shh, and Notch[18]. Indeed, the expression of a constitutively active Fgf3 mutant allele (K664E) in mice leads to an increase in cortical thickness, but not cortex folding[19]. Ventricular injection of FGF2 leads to increased tangential growth of the VZ and proliferation of IP that correlates with cortex folding[20]. Similarly, local induction of FGF8 in the cortex increases the number of IPs and the formation of additional gyri in the ferret[21]. Conversely, suppressing all FGFR activity in the ferret cortex impairs the formation of cortical folds[22]. Ablation of Fgf2 in mice leads to reduced expansion and thickening of the cortex[23], but this phenotype is opposite to that in Fgf10 knockout mice[24]. FGF10 is expressed mainly by rostral cortical NECs during embryonic day 9 (E9) to E11, where it supports their differentiation into aRGs[25]. Genetic deletion of Fgf10 delays their differentiation, favoring their tangential amplification. This results in an increased number of aRGs and neurons, leading to the expansion and thickening of the cortex in rostral regions[24].

Besides proliferation, neuronal migration is another key mechanism that drives cortex folding. Indeed, the role of several genes linked to abnormal folding in humans has been associated with neuronal migration in ferret and mouse models[4,26]. In gyrencephalic species such as the ferret, migrating neurons do not follow strictly parallel pathways as observed in the mouse cortex[27], but instead, follow divergent trajectories concomitant with the onset of cortical folding[28]. This is consistent with the observation that neurons in ferret cortices can switch between radial fibers and enhance their lateral dispersion, compared to those in the mouse[29]. We provided a molecular mechanism for the role of neuronal migration in cortex folding, demonstrating that genetic knockout of Flrt1/3 adhesion molecules leads to faster and divergent migration, as well as sulcus formation, in the mouse cortex without changes in cell proliferation[30]. This phenotype arises from changes in the intercellular adhesion among migrating neurons, where approximately 50% are Flrt-mutant neurons (previously destined to become Flrt+ in wild-type animals). These Flrt-mutant neurons exhibit reduced intercellular adhesion, migrate faster, and segregate from their neighboring Flrt-negative neurons. A similar scenario has been observed for the Eph/ephrin family of guidance receptors/ligands. Overexpression of ephrinB1, which can induce homophilic cell adhesion[31], reduces the horizontal dispersion of multipolar neurons[32]. Likewise, EphA/ephrinA gain-of-function experiments show reduced lateral dispersion of multipolar neurons[33]. Similarly, ephrinA2/A3/A5 triple knockout (tKO) mice, like the Flrt1/3 dKO phenotype, display neuronal segregation along the tangential axis, leading to a wavy cortical plate with alternating thicker and thinner areas[33].

A widely accepted hypothesis suggests that cortex folding is driven by a combination of uneven progenitor expansion and divergent radial migration[34]. Yet, we still do not fully understand the logic behind how coordinated and simultaneous events like proliferation and divergent migration lead to the formation of sulci and gyri in the cortex. Here, we addressed this question by taking advantage of the Flrt1/3 dKO mouse line, which promotes divergent migration, and combined it with other genetic lines that induce progenitor expansion. We chose two different lines that expand either intermediate or apical progenitors through genetic ablation of Cep83 and Fgf10, respectively. This allowed us to dissect the specific contributions of distinct progenitor pools to cortex folding when coupled with increased divergent migration. We found that expansion of intermediate progenitors favors sulcus formation, whereas amplification of apical progenitors promotes gyrus formation in the Flrt1/3 dKO model. Single cell transcriptomics and simulations suggest that changes in adhesive properties of cortical neurons, cell sorting by differential adhesion, increased cell densities in the cortical plate, combined with lateral dispersion during their radial migration are important folding parameters. In this work, our results reveal that the combination of increased proliferation and divergent cell migration enhances their effect on cortex folding beyond the individual effects. Moreover, we show that the formation of gyri involves slightly but significantly different mechanisms from the formation of sulci.

## Results

### Cep83 and Flrt1/Flrt3 loss enhances sulci-like cortex folding

To investigate the contributions of progenitor cell expansion and divergent cell migration to cortex folding, we combined the deletion of Cep83 with the double deletion of Flrt1 and Flrt3. In previous work[30] we had used the pan-nervous system Nestin-Cre driver to conditionally delete Flrt3 and, together with the Flrt1 null allele, to generate Flrt1/3 double knockout (dKO) mice. The conditional deletion of Cep83 was previously done with the Emx1-Cre driver[17] which is expressed predominantly in the developing cortex and hippocampus starting at E10.5 with an earlier developmental time course than Nestin-Cre (Supplementary Fig. 1a, b). We decided to use the Emx1-Cre driver to generate Cep83;Flrt1/3 triple KO mice (from now on referred to as Cep83 tKO mice). We validated that the Emx1-Cre driver was effective in deleting Cep83 and Flrt3 protein expression (Supplementary Fig. 1c, d) and that the phenotype of Emx1-Cre;Flrt1/3 dKO mice (from now on referred to as Flrt1/3 dKO mice) was overall very similar to Nestin-Cre;Flrt1/3 dKO mice. At E17.5, the size of the Flrt1/3 dKO cortex was not enlarged compared to Flrt1−/−Flrt3lx/+ mice or heterozygous controls, consistent with the previous observation that progenitor populations were not expanded (Fig. 1a; Supplementary Fig. 1e and Ref. 30). Approx. one third of the animals had folds, often only a single fold, and typically in the lateral cortex (Fig. 1b). Occasionally, we also observed folds in the lateral region of the cortex in control littermates, deficient for Flrt1, in line with previous findings[30] (Supplementary Fig. 1f). We also analyzed cortex size and folding of E17.5 Emx1-Cre;Cep83 single cKO embryos, and found that the cortex was expanded as expected (Fig. 1c; Supplementary Fig. 1g) and folds were observed in the medial (cingulate) cortex as previously described for P21 mutants[17] (Supplementary Fig. 1h). We then generated Cep83 tKO mice and found that cortex sizes of E17.5 embryos were enlarged compared to controls (Fig. 1d, e). Interestingly, the number of cortical folds in Cep83 tKO mice was more numerous than in the Cep83 single and Flrt1/3 dKO mice

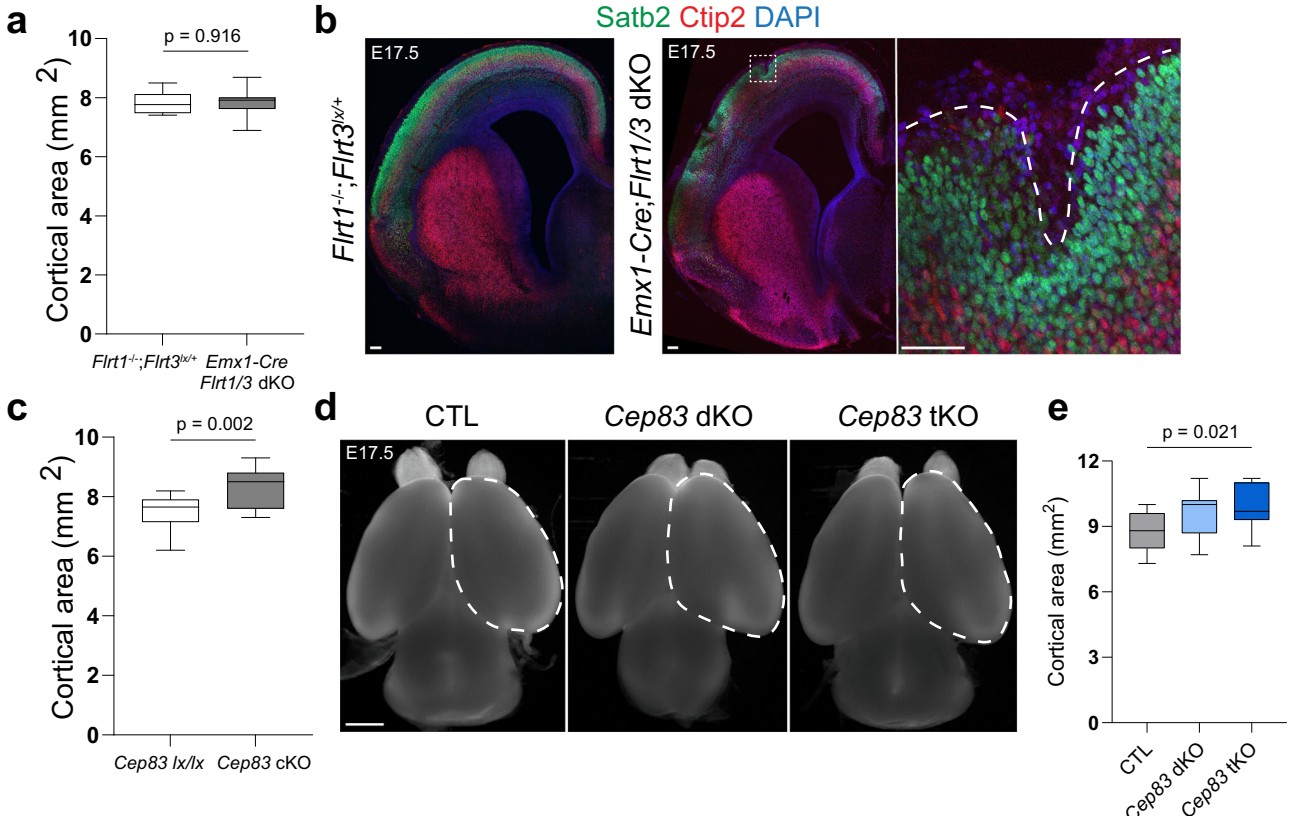

**Fig. 1 | Cortex enlargement induced by Cep83 and Flrt1/Flrt3 deletion.**
**a** Quantification of cortical areas. Data are represented as a box plot, with median (center line), interquartile range (box) and minimum and maximum values (whiskers); *Flrt1* $^{-/-}$*Flrt3* $^{lx/+}$, $n = 9$ brains; *Emx1-Cre;Flrt1/3* dKO, $n = 13$ brains from 7 litters, two-tailed t-test with Welch correction. **b** E17.5 *Flrt1* $^{-/-}$*Flrt3* $^{lx/+}$ and *Emx1-Cre;Flrt1/3* dKO brain sections labeled with Satb2 (green), Ctip2 (red), and DAPI (blue). Area in dashed rectangle is shown with higher magnification on the right. Scale bars,

100 μm. **c** Quantification of E17.5 cortical areas. *Cep83* lx/lx, $n = 14$ brains; *Cep83* cKO, $n = 16$ brains from 4 litters. Data are represented as a box plot (as above). **d** Representative whole-mount images of E17.5 CTL, *Cep83* dKO, and *Cep83* tKO brains. Dashed areas were measured to obtain quantification in (**e**). Scale bar, 1 mm. **e** Quantification of cortical areas. CTL, $n = 11$ brains; *Cep83* dKO, $n = 13$ brains; *Cep83* tKO $n = 19$ brains from 10 litters. Data are represented as a box plot (as above).

combined. Typically, cortical folds could be seen in freshly dissected brains and after tissue clearing and light sheet imaging (Fig. 2a, b; Supplementary Movie 1, 2). Folds were often seen as individual sulci in lateral and/or cingulate cortex, either in one or both cerebral hemispheres. (Fig. 2c–e). Cortical layers seemed preserved as shown by Satb2 immunostainings, a marker for upper layer neurons (and callosally projecting neurons), and Ctip2, a marker for lower layer neurons (Fig. 2c–e; Supplementary Fig. 1i, j). The apical surface of the VZ remained unfolded, indicating bona fide foldings[6] (Fig. 2c–e). Importantly, the penetrance of the phenotype was greatly increased from approx. one-third in *Cep83* single and *Flrt1/3* dKO mice to over 80% penetrance in *Cep83* tKO mice (Fig. 2f, g). Remarkably, mice homozygous for Cep83 and Flrt1 deletion, but heterozygous for Flrt3 (*Emx1-Cre;Cep83*$^{fl/fl}$*;Flrt1*$^{-/-}$*;Flrt3*$^{fl/+}$ double KO (*Cep83* dKO)) also showed increased penetrance of cortical folding (Fig. 2f, g). This was consistent with previous observations in Nestin-Cre driven *Flrt1/3* mutant mice[30]. To analyze if there was synergy in the folding phenotype, we calculated the number of folds in each brain of the relevant cohorts and asked if the value of the triple knockouts was significantly larger than that expected from a mere additive effect. To calculate the additive effect, we summed up the average number of folds in *Cep83* single and *Flrt1/3* dKO corrected by the average number of folds in *Cep83* lx/lx controls. This value summed up to 0.6 (Fig. 2h). The comparison with the number of folds in the combined *Cep83* dKO and *Cep83* tKO cohorts revealed that the phenotypes in the combined lines were significantly greater than the theoretical additive effect (Fig. 2h). We calculated the gyrification index (GI) (a measure of the real length of the cortical

surface over a hypothetical minimal length of the cortical surface) and found that in rostral and medial regions of the *Cep83* tKO cortex, the GI was significantly higher than in single *Cep83* cKO or *Flrt1/3* dKO mice (Fig. 2i, j). These results indicate that deletion of Cep83 together with deletion of Flrt1/3 leads to a strong cortex folding phenotype in both cerebral hemispheres.

### Increased intermediate progenitors in *Cep83* tKO mice

We next investigated the underlying mechanism of the enhanced folding phenotype of *Cep83* tKO mice. Previous work had indicated an increased density of Tbr2+ IPs in the subventricular zone of *Cep83* cKO embryos, whereas the density of mitotic stem cells positive for the marker phosphorylated histone H3 (PH3) at the ventricular zone surface had remained unchanged[17]. We therefore asked if the deletion of Flrt1/3 in addition to Cep83 would alter the spectrum of progenitors in the developing cortex. We performed quantifications in *Cep83* tKO embryos of apical progenitors (Sox2 + ), IPs (Tbr2 + ), mitotic dividing cells (PH3 + ), Pax6/PH3-double positive cells, and Pvim+ cells with a basal process located outside the VZ[35] (Fig. 3; Supplementary Fig. 2). We used *Flrt1*$^{-/-}$*;Flrt3*$^{lx/+}$ embryos as controls, because they could be generated as littermates and were previously shown to have normal numbers of cortical progenitors[30]. At both rostral and medial locations, we observed significantly increased cell numbers of basal IPs (Tbr2 + ) and Pax6/PH3-double positive cells compared to controls, consistent with previous findings in *Cep83* single KO[17] (Fig. 3). We also observed more Pvim+ cells with a basal process in *Cep83* tKO embryos (Supplementary Fig. 2G). The total number of apical progenitors

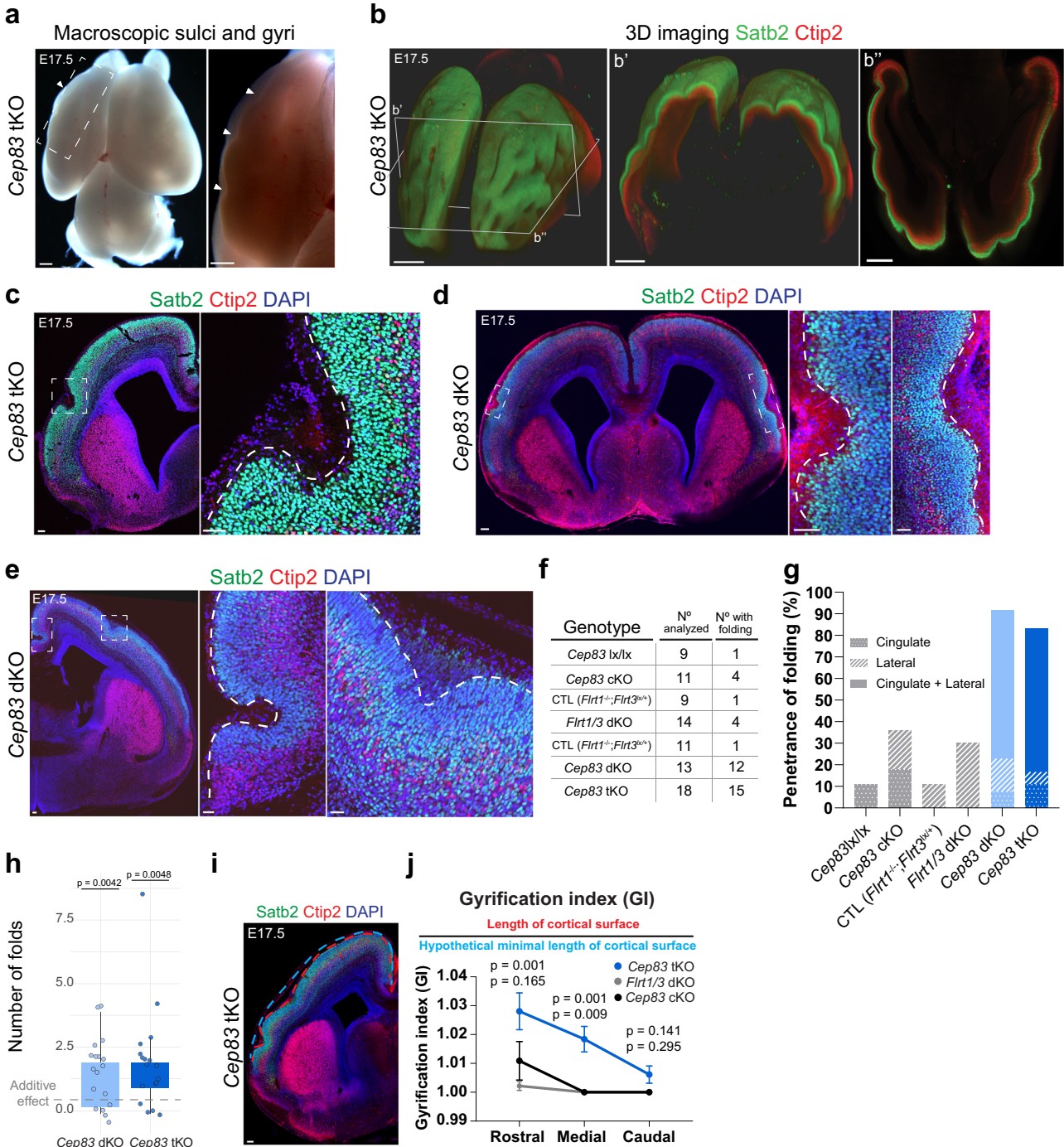

**Fig. 2 | Loss of Cep83 and Flrt1/Flrt3 enhances sulci-like cortex folding.**
**a** Macroscopic sulci and gyri in an E17.5 *Cep83* tKO embryo. The dashed rectangle is shown with higher magnification on the right; sulci and gyri are indicated by arrowheads. Scale bars 500 µm. **b-b"** 3D imaging of the E17.5 *Cep83* tKO embryo shown in **a** labeled with Satb2 (green) and Ctip2 (red). 3D whole brain, XY and YZ planes are shown (see Supplementary Movie 1,2). Scale bars, 300 µm, 400 µm, 400 µm. **c−e** Sections from E17.5 *Cep83* tKO (**c**) and *Cep83* dKO (**d, e**) brains labeled with Satb2 (green), Ctip2 (red), and DAPI (blue). Dashed rectangles: one sulcus (**c**), bilateral sulci (**d**), two sulci in cingulate and lateral cortex (**e**); higher magnifications are shown on the right. Scale bars, **c** 100 µm, 50 µm. **d** 200 µm, 100 µm. 100 µm. **e** 100 µm, 50 µm, 50 µm. **f, g** Folding penetrance at E17.5 *Cep83* lx/lx, *Cep83* cKO, CTL (*Flrt1⁻/⁻;Flrt3*^lx/+; littermate controls of *Flrt1/3* dKO and *Cep83* tKO), *Flrt1/3* dKO, *Cep83* dKO, and *Cep83* tKO mice. Sulcus locations indicated in the graph: Cingulate

cortex (dotted), lateral cortex (diagonal), both (solid). **h** Statistical analysis of synergistic vs additive effects in the number of folds between *Cep83* single and *Flrt1/3* dKO vs *Cep83* dKO or *Cep83* tKO animals. The additive effect was calculated as the sum of mean cortical folds in *Cep83* cKO and *Flrt1/3* dKO, corrected for the background of cortical folds in control *Cep83* lx/lx animals. Box plots (as in Fig. 1). one-sample Wilcoxon test with one-sided alternative. **i** Representative immunostained image of a *Cep83* tKO embryo depicting the quantification of the Gyrification Index (GI). Red line: de facto cortical surface length; blue line: minimal hypothetical length. Scale bar, 100 µm. **j** GI quantification. *Cep83* single cKO (black) *n* = 11 brains/ 3 litters, *Flrt1/3* dKO (gray) *n* = 13 brains/7 litters, *Cep83* tKO (blue) *n* = 31 sections from 18 brains/10 litters, quantified at rostral, medial, and caudal levels. Data are shown as mean ± SEM, one-way ANOVA with Tukey's post hoc analysis.

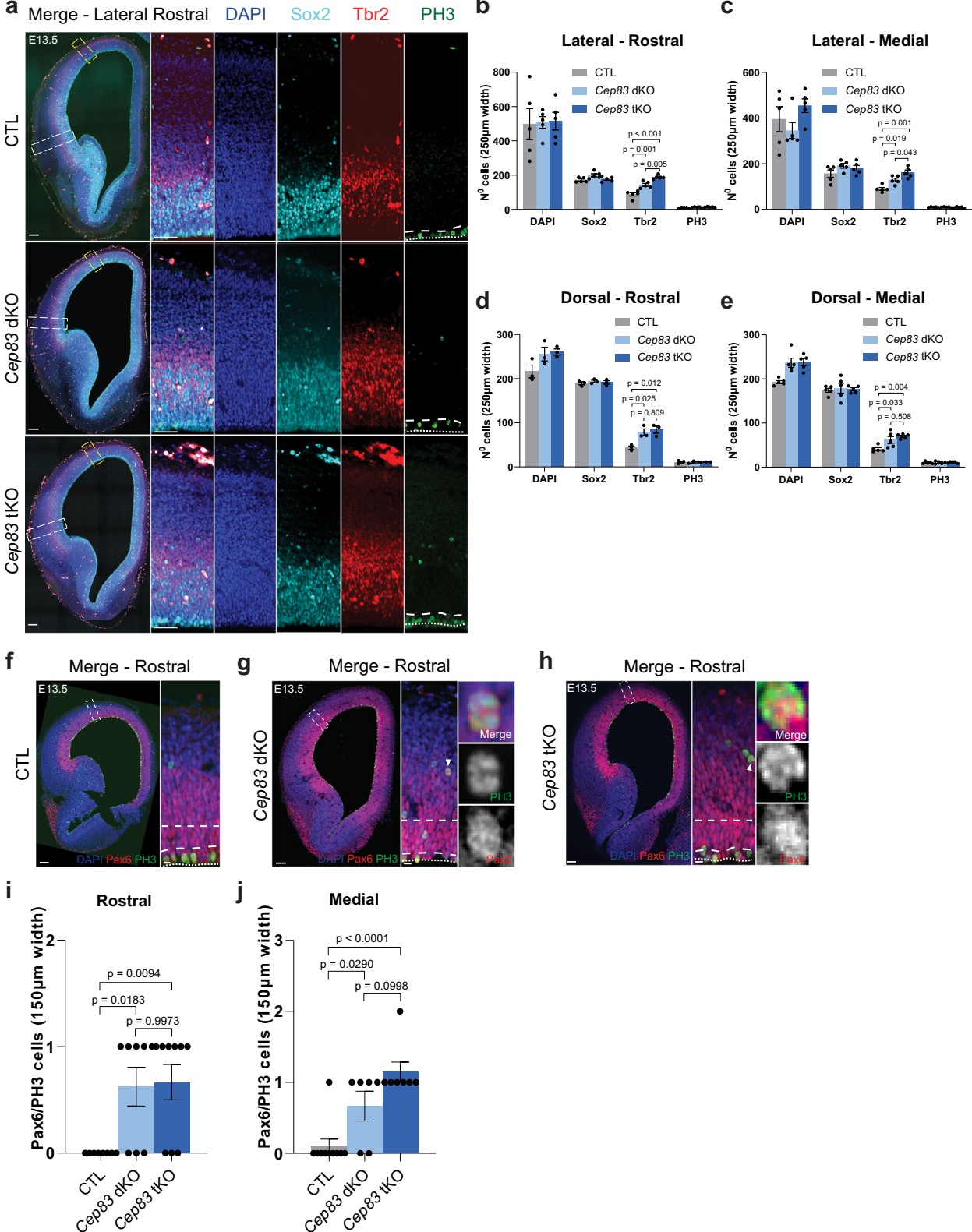

(Sox2+) and mitotic dividing cells remained unchanged (Fig. 3a–e, Supplementary Fig. 2a, d, e). Similar results were obtained with *Cep83* dKO embryos, consistent with the observed folding phenotype (Fig. 3a–j). Collectively, these results suggest that deletion of Cep83 causes an expansion of IPs and Pax6/PH3-double positive cells at E13.5 that results in an enlarged cortex at E17.5. Since this effect is already present in *Cep83* cKO mice and is only mildly enhanced, if at all, by the

deletion of Flrt1 and Flrt3, it follows that the expansion of IPs and Pax6/PH3-double positive cells in *Cep83* tKO embryos exacerbates the cortex folding phenotype seen in the *Flrt1/3* dKO mice.

### Fgf10 and Flrt1/Flrt3 loss enhances gyri-like cortex folding
We next investigated the interaction of progenitor expansion and cortical cell migration in a separate mouse model, this time involving

**Fig. 3 | Increased intermediate progenitors in *Cep83* tKO and Cep83 dKO mice.**
**a** Rostral cortices from E13.5 control (CTL), *Cep83* dKO and tKO embryos stained with DAPI (blue), Sox2 (cyan, early progenitors), Tbr2 (red, intermediate progenitors), and PH3 (green, mitotic cells). Apical VZ border: dotted line, basal side: dashed line in PH3 images. White dashed rectangles (lateral-rostral) are shown with higher magnification on the right. Yellow dashed rectangles indicate–dorsal-rostral areas. Scale bars, 100 μm, 50 μm. **b**, **c** Quantifications of cell densities in lateral regions shown in (**a**) for rostral (**b**) and medial (**c**, Supplementary Fig. 2a) areas. (CTL *n* = 5 brains, *Cep83* dKO *n* = 5 brains, *Cep83* tKO *n* = 5 brains from 3 litters). Data are shown as mean ± SEM, one-way ANOVA with Tukey's post hoc analysis. **d**, **e** Additional quantification of dorsal areas in rostral (**d**, Supplementary Fig. 2d) (CTL *n* = 3 brains, *Cep83* dKO *n* = 3 brains, *Cep83* tKO *n* = 3 brains, 2 litters); and medial

(**e**, Supplementary Fig. 2e). (CTL *n* = 5 brains, *Cep83* dKO *n* = 5 brains, *Cep83* tKO *n* = 5 brains, 3 litters). Data are shown as mean ± SEM, one-way ANOVA with Tukey's post hoc analysis. **f–h** E13.5 rostral cortices from CTL (**f**), *Cep83* dKO (**g**), and tKO (**h**) stained with DAPI (blue), Pax6 (red, RG cells), and PH3 (green). Apical VZ border: dotted line, basal side: dashed line. Arrowheads indicate Pax6/PH3+ cells located >60 μm above the apical surface (outside the VZ); high magnifications shown on the right. Scale bars, **f–h** 100 μm, 10 μm. **i** Quantification of Pax6/PH3+ cells from (**f–h**). (CTL *n* = 8 brains, *Cep83* dKO *n* = 8 brains, *Cep83* tKO *n* = 9 brains from 5 litters). Data are shown as mean ± SEM, one-way ANOVA with Tukey's post hoc analysis. **j** Similar to (**i**), but from data shown in Supplementary Fig. 2f. (CTL *n* = 10 brains, *Cep83* dKO *n* = 6 brains, *Cep83* tKO *n* = 7 brains from 5 litters). Data are shown as mean ± SEM, one-way ANOVA with Tukey's post hoc analysis.

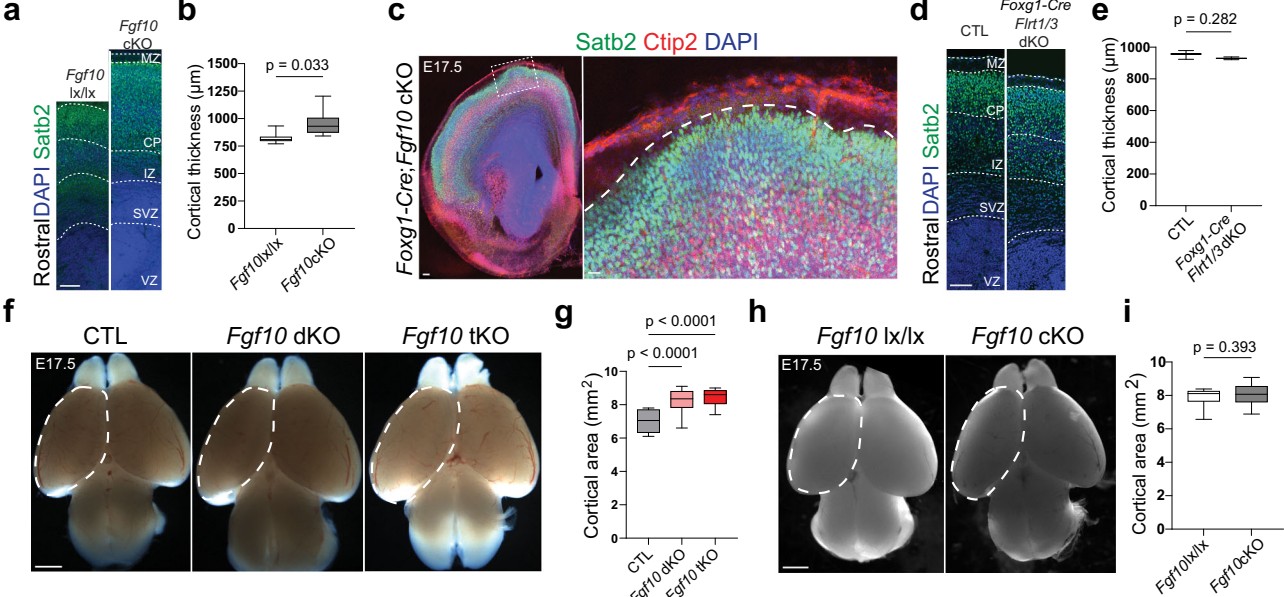

**Fig. 4 | Cortex enlargement induced by Fgf10 and Flrt1/Flrt3 deletion.**
**a** Representative Satb2 (green) and DAPI (blue) stained E17.5 rostral sections of *Fgf10* lx/lx control and *Fgf10* cKO mice. VZ ventricular zone, SVZ subventricular zone, IZ intermediate zone, CP cortical plate, MZ marginal zone. Scale bar, 100 μm. **b** Quantifications of cortical laminar thickness shown in (**a**). Measurements were obtained from the bottom of the apical surface of the VZ to the pial surface. *Fgf10* lx/lx, *n* = 6 brains; *Fgf10* cKO, *n* = 6 cortices from 3 brains from 4 litters. Data are represented as a box plot (as in Fig. 1). two-tailed t-test with Welch correction. **c** Representative E17.5 Foxg1-Cre;*Fgf10* cKO brain section labeled with Satb2 (green), Ctip2 (red), and DAPI (blue). Area in dashed rectangle is shown with higher magnification on the right. Dashed line indicates a gyrus. Scale bars, 200 μm. **d** Representative Satb2 (green) and DAPI (blue) stained rostral sections of CTL and *Flrt1/3* dKO mice. Scale bar, 100 μm. **e** Quantifications of cortical laminar thickness

shown in (**d**). Measurements were taken as in (**b**). CTL *n* = 3 brains; *Flrt1/3* dKO, *n* = 3 brains from 3 litters. Data are represented as a box plot (as in Fig. 1). two-tailed t-test with Welch correction. **f** Representative whole-mount images of E17.5 CTL, *Fgf10* dKO and *Fgf10* tKO brains. Dashed areas were measured to obtain quantifications in (**g**). Scale bar, 1 mm. **g** Quantifications of the cortical areas shown in (**f**). CTL, *n* = 8 brains; *Fgf10* dKO, n = 22 brains; *Fgf10* tKO, n = 13 brains from 11 litters. Data are represented as a box plot (as in Fig. 1). one-way ANOVA, Tukey Post hoc test. **h** Representative whole-mount images of E17.5 *Fgf10* lx/lx and *Fgf10* cKO brains. Dashed areas were measured to obtain quantifications in (**i**). Scale bar, 1 mm. **i** Quantifications of the cortical areas shown in (**h**). *Fgf10* lx/lx, *n* = 21 brains; *Fgf10* cKO, *n* = 24 brains from 5 litters. Data are represented as a box plot (as in Fig. 1). two-tailed t-test with Welch correction.

the deletion of Fgf10. *Fgf10* null animals displayed delayed RG differentiation causing enhanced symmetric divisions of NECs early during cortical development eventually resulting in an enlargement of cortical areas at birth[24]. The cortical expansions were restricted to rostral regions, because Fgf10 expression was confined to the rostral cortex. To generate conditional *Fgf10;Flrt1/3* tKO mice (in short *Fgf10* tKO mice), we used the Foxg1-Cre driver, which was previously used in combination with a Fgf10[lx/lx] allele to enlarge the brain (D. Kawaguchi, Y. Gotoh, unpublished results). Foxg1-Cre expresses uniformly in telencephalic progenitors as early as E9.0-E9.5[36], one day earlier than Emx1-Cre using the Cre-dependent reporter pCALNL-TdTom (Supplementary Fig. 3a). We also confirmed that the Foxg1-Cre driver deleted expression of Fgf10 and Flrt3 in the cortex (Supplementary Fig. 3b-e). We next asked if *Foxg1-Cre* mice could be used to generate *Flrt1/3* dKO mice. Indeed, we found that *Foxg1-Cre;Flrt1/3* dKO mice displayed a cortex folding phenotype that was qualitatively similar to

*Nestin-Cre;Flrt1/3* dKO mice (Supplementary Fig. 3f), consisting mostly of a single sulcus, albeit with very low penetrance (see below). Next, we investigated the effects of Fgf10 deletion on cortex size. As expected, we found that the absence of Fgf10 increased rostral cortical thickness compared to controls lacking Foxg1-Cre (Fig. 4a, b). In few embryos, this phenotype correlated with a gyrus-like protrusion (Fig. 4c). This effect was not observed in *Foxg1-Cre;Flrt1/3* dKO mice with an intact Fgf10 gene (Fig. 4d, e). Next, we generated *Fgf10* tKO mice and, by measuring the entire size of the cortex, found that it was expanded (Fig. 4f, g). This was unexpected, since using the same measurements, neither *Fgf10* single cKO nor *Flrt1/3* dKO mice had an expanded cortex (Fig. 4h, i; Supplementary Fig. 3g, h).

When analyzing cortex folding, we found that *Fgf10* tKO mice showed enhanced cortex folding in both cerebral hemispheres, but unlike *Cep83* tKO mice, folds more often resembled gyri (Fig. 5a–c). Overall, cortical layers were preserved as indicated by the markers

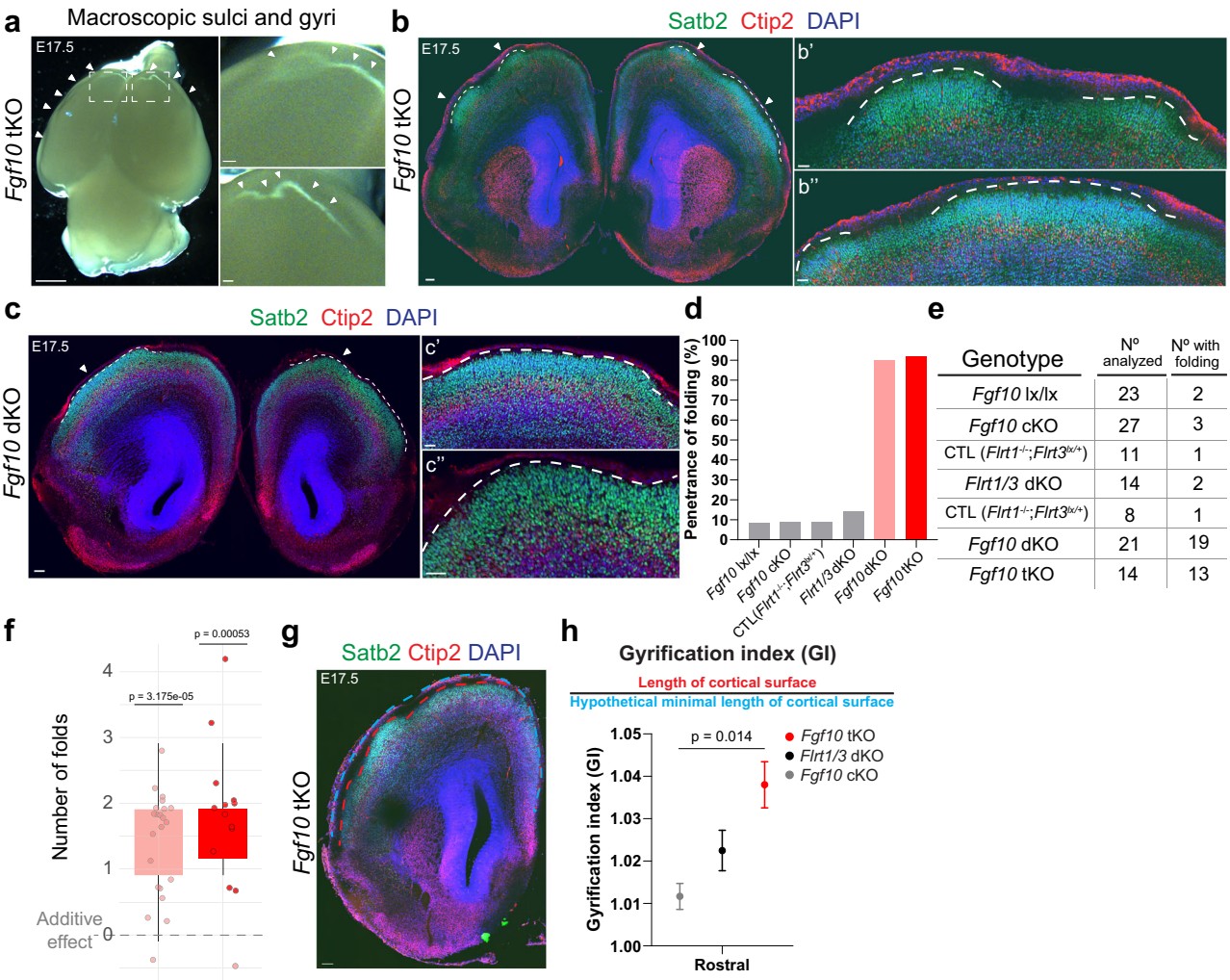

**Fig. 5 | Fgf10 and Flrt1/Flrt3 loss enhances gyri-like cortex folding.**
**a** Macroscopic sulci and gyri in E17.5 *Fgf10* tKO embryo. Dashed rectangles are shown with higher magnification on the right, sulci and gyri are indicated by arrowheads. Scale bars, 500 µm. **b-b″** E17.5 *Fgf10* tKO brain. **c-c″** E17.5 *Fgf10* dKO brain stained with Satb2, Ctip2, and DAPI. Dashed lines indicate gyri in both hemispheres. Higher magnifications on the right (**b′**, **b″**, **c′**, **c″**). Scale bars, **b**, **c** 100 µm, **b′-b″** and **c′-c″** 50 µm. **d** Folding penetrance of E17.5 *Fgf10* lx/lx, *Fgf10* cKO, CTL (Flrt1−/−;Flrt3lx/+), *Flrt1/3* dKO, *Fgf10* dKO, and *Fgf10* tKO. All folds located on the lateral cortex. **e** Table of cortical folding penetrance at E17.5. Brains were analyzed for the presence of one or more gyri, as in (**d**). *Fgf10* lx/lx embryos were littermate controls for the *Fgf10* cKO. To increase the number of *Fgf10* cKO embryos, mothers were *Foxg1-Cre;Fgf10* lx/lx. Germline Cre activity may explain folding in two *Fgf10*

lx/lx embryos. **f** Statistical analysis of synergistic vs additive effects on number of folds in *Fgf10* single and *Flrt1/3* dKO vs *Fgf10* dKO or tKO. Additive effect: sum of mean cortical folds in *Fgf10* single and *Flrt1/3* dKO, corrected for background in *Fgf10* lx/lx controls. Data are represented as a box plot (as in Fig. 1). one-sample Wilcoxon test with one-sided alternative. **g** Representative image of a *Fgf10* tKO embryo depicting the quantification of the Gyrification Index (GI). Red line: de facto cortical surface; blue line: minimal (hypothetical) length. Scale bar, 100 µm. **h** GI quantification in E17.5 embryos. n = 6 brains/3 litters *Fgf10* cKO (gray), n = 4 brains/3 litters *Flrt1/3* dKO (black) and n = 15 sections from 7 brains/5 litters *Fgf10* tKO (red) in the rostral cortex. Data are shown as mean ± SEM. one-way ANOVA with Tukey's post hoc analysis.

Satb2 and Ctip2, and the apical surface of the VZ remained unaltered, indicating a bona fide cortical folding phenotype[6] (Fig. 5b, c; Supplementary Fig. 3i, j). Penetrance was very much enhanced from approximately. 10 to 15% in *Fgf10* single cKO and *Flrt1/3* dKO mice to over 90% penetrance in *Fgf10* tKO mice (Fig. 5d, e). The folding phenotype was also highly penetrant in mice homozygous for Fgf10 and Flrt1 deletion, and heterozygous for Flrt3 (*Foxg1-Cre;Fgf10*fl/fl;*Flrt1−/−;Flrt3*fl/+ double KO [in short *Fgf10* dKO]) (Fig. 5d, e), consistent with the *Cep83* dKO situation). To analyze if there was synergy in the folding phenotype, we performed a similar analysis as for the *Cep83* mutant mice. We calculated the number of gyri in each brain of the relevant cohorts and asked if the value of the triple knockouts was significantly larger than that expected from a mere additive effect. The comparison revealed that the phenotypes in the combined lines were significantly greater than the theoretical additive effect (Fig. 5f). The GI was significantly higher in *Fgf10* tKO than *Fgf10* single cKO mice (Fig. 5g, h).

These results indicate that deletion of Fgf10, combined with Flrt1/3 ablation, leads to a stronger cortex folding phenotype in both cerebral hemispheres.

**Increased numbers of apical progenitors in *Fgf10* tKO mice**
Ablation of Fgf10 was previously shown to expand apical progenitors, eventually resulting in an overproduction of neurons in the rostral cortex[24]. To investigate the underlying mechanism of the enhanced folding phenotype in *Fgf10* tKO mice, we asked if the deletion of Flrt1/3 in addition to Fgf10 would alter other types of progenitors in the developing cortex. We performed quantifications of apical progenitors (Sox2), IPs (Tbr2, Supplementary Fig. 4g, h), and mitotic dividing cells (PH3) in rostral and intermediate cortical sections of E11.5 *Fgf10* tKO embryos. As expected, we found that Sox2-positive apical progenitors were expanded in rostral sections of *Fgf10* dKO and tKO mice compared to littermate controls (*Flrt1−/−;Flrt3*lx/+) (Fig. 6a–c; Supplementary

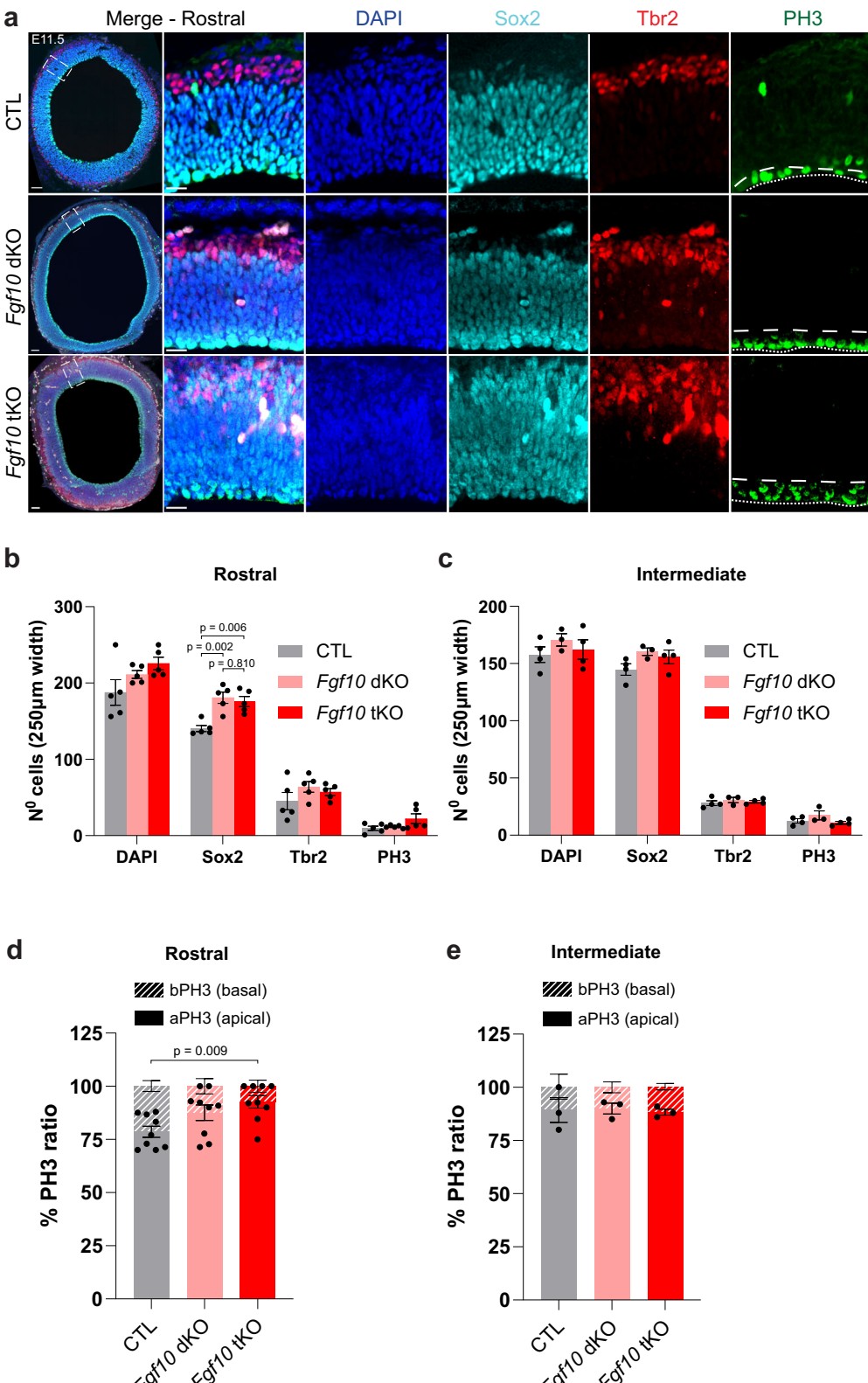

Fig. 4d). The numbers of Tbr2-positive IPs and Pax6/PH3-double positive cells were not increased in E11.5 or E13.5 *Fgf10* dKO and tKO embryos (Fig. 6a–c; Supplementary Fig. 4a–f). At E11.5, the proportions of mitotic cells (PH3 + ) between apical versus basal layers were shifted towards apical layers in rostral sections of *Fgf10* dKO and tKO embryos (Fig. 6d, e), suggesting increases in dividing NECs and aRG cells. A somewhat weaker effect could still be seen in E13.5 embryos,

consistent with Fgf10's described early and transient effects[24] (Supplementary Fig. 4c). Together, these results suggest that the deletion of Flrt1/3 in addition to Fgf10 did not alter other types of progenitors compared to the *Fgf10* single cKO mice. It follows that the expansion of APs in *Fgf10* tKO embryos exacerbates the cortex folding phenotype seen in the *Flrt1/3* dKO mice. Together with the findings in *Cep83* tKO mice, these results indicate that expansion of cortical progenitors in

**Fig. 6 | Progenitor expansion in *Fgf10* tKO *and Fgf10* dKO embryos at E11.5.**
**a** E11.5 rostral cortices of CTL, *Fgf10* dKO and *Fgf10* tKO embryos were stained DAPI (blue), apical progenitors Sox2 (cyan), intermediate progenitors Tbr2 (red), and mitotic cells PH3 (green). Apical and basal sides of the VZ are indicated with dotty and dashed lines in PH3 stained images, respectively. Areas in dashed rectangles in (**a**) are shown with higher magnification on the right. Scale bars, 50 μm, 20 μm. **b** Quantification of the data shown in (**a**). CTL *n* = 5 brains, *Fgf10* dKO *n* = 5 brains, *Fgf10* tKO *n* = 5 brains from 4 litters. Data are shown as mean ± SEM. one-way ANOVA with Tukey's post hoc analysis. **c** Quantification of the data shown in

Supplementary Fig. 4d. CTL *n* = 4 brains, *Fgf10* dKO *n* = 3 brains, *Fgf10* tKO *n* = 4 brains from 4 litters. Data are shown as mean ± SEM; no significant changes between groups, one-way ANOVA with Tukey's post hoc analysis. **d** Proportion of apical/basal mitotic cells (PH3) in rostral region, CTL n = 9 brains, *Fgf10* dKO n = 9 brains, *Fgf10* tKO *n* = 9 brains from 7 litters. Data are shown as mean ± SEM. one-way ANOVA with Tukey's post hoc analysis. **e** Proportion of apical/basal mitotic cells (PH3) in intermediate region, CTL *n* = 3 brains, *Fgf10* dKO *n* = 3 brains, *Fgf10* tKO *n* = 3 brains from 3 litters. Data are shown as mean ± SEM; no significant changes between groups, one-way ANOVA with Tukey's post hoc analysis.

---

conjunction with alteration of cell migration drives folding of the mouse cortex. The types of folds (sulci versus gyri) appear to depend on the types of progenitors that were expanded: expansion of apical progenitors in *Fgf10* tKO mice correlated with the appearance of gyri, while expansion of IPs and Pax6/PH3-double positive cells in *Cep83* tKO mice correlated with the appearance of sulci (Supplementary Fig. 4i).

### Higher cell density in gyri of *Fgf10* tKO mice

To further characterize the cortical folds in both tKO mouse models, we quantified and compared the cell densities within the folded areas. Naturally forming gyri in gyrencephalic species tend to have a higher cell density than sulcus areas[3,7,37,38]. It was therefore interesting to ask if the generated gyri in *Fgf10* tKO mice would have higher cell densities than control areas in the same cortical region. We quantified the densities of Satb2-positive upper-layer cortical neurons and found that the cell densities in the gyrus regions were approximately 50% higher than those in adjacent areas or in an equivalent region of a littermate control brain (Fig. 7a–c). In contrast, the densities of Satb2-positive neurons in the sulcus regions of *Cep83* tKO mice were on average similar to those in other regions of the same section or control brains (Fig. 7d–f). Conversely, the number of Ctip2-positive lower-layer neurons was reduced in the sulcus regions of *Cep83* tKO mice, but not in the gyrus regions of *Fgf10* tKO mice, compared to adjacent areas (Supplementary Fig. 5a–c). This is consistent with a thinning of their layer in sulcus but not gyrus regions, as found in gyrencephalic species[39]. No changes in apoptosis were observed in either region for both models (Supplementary Fig. 5d–g), and radial glia fibers showed the classical trajectory of convergence in sulcal pits for *Cep83* tKO and divergence in gyrus regions for *Fgf10* tKO (Supplementary Fig. 5h, i). Importantly, cortex folding was also found in postnatal (P1) stages in both models (sulci-like in case of *Cep83* tKO and gyri-like in case of *Fgf10* tKO mice), and the penetrance was high (60% for *Cep83* tKO and 80% for *Fgf10* tKO) (Fig. 7g–j). These findings suggest that folding in *Fgf10* tKO and *Cep83* tKO mice partially recapitulate features of physiological cortical folds.

### Increased neuronal migration speed in *Cep83* tKO and *Fgf10* tKO mice

We had previously reported that loss of Flrt1/3 resulted in cortical layer thinning in sulcus areas, particularly in lower CP, similar to gyrencephalic species such as the ferret[39] and primates[40]. We found a similar reduction in the thickness of lower layers in the folded areas of *Cep83* tKO mice, but not in the *Fgf10* tKO mice, compared to controls (*Flrt1⁻/⁻Flrt3^{lx/+}*) (Fig. 8a, b; Supplementary Fig. 5a–c). This is consistent with the *Cep83* tKO favoring sulcus and the *Fgf10* tKO favoring gyrus formation. The proportion of Satb2+ cells in upper and lower CP shifted towards upper layer CP in both mutants (Fig. 8c, d), consistent with previous observations after ablation of Flrt1/3[30] and with mutant pyramidal neurons migrating faster through the cortical plate. Support for such a model came from bromodeoxyuridine (BrdU) pulse labeling of newborn pyramidal neurons at E13.5 and analyzing their distribution in the CP at E16.5. In both tKO models, we observed a shift in the proportion of BrdU+ neurons in the upper versus lower CP compared

to their respective controls, and this shift occurred throughout the CP independent of folded areas (Fig. 8e–h).

Previously, we had used ex vivo live imaging of embryonic cortices to demonstrate that loss of Flrt1/3 increases neuronal migration speed[30]. We took the same approach to analyze the speed of radial migration in both tKO models, although these experiments were extremely challenging due to the low yield of in utero electroporation (IUE) of tKO embryos. IUE with pCAG-Ires-GFP was performed at E13.5 and time-lapse imaging was done for up to 48 h from E15.5 cortical slices (Supplementary Fig. 5j–l). Migrating GFP-labelled neurons were tracked within the CP and the average speeds were calculated. The results suggested that the average speeds of *Cep83* tKO and *Fgf10* tKO neurons were faster than control neurons (Supplementary Fig. 5m, n). Together, these results suggest that the enhanced cortical folding phenotype in the triple KO mice is the result of combining progenitor expansion with faster migration.

### Modeling cortex folding in *Cep83* tKO and *Fgf10* tKO mice

Previously, we had used data-driven computational simulations to provide evidence that cell clustering and increased migration speed produced cortical folds[30]. To model the combined effects of progenitor expansion and cell migration on cortex folding in the tKO mice, we first analyzed the global cellular changes in the tKO mice by scRNAseq. We dissected the rostral and medial parts of the neocortex from three E15.5 *Cep83* tKO and three *Fgf10* tKO embryos plus respective controls and performed scRNAseq to identify the different cortical cell types (Fig. 9a). We obtained 19268 and 14777 single cell transcriptomes from *Cep83* tKO and *Fgf10* tKO embryos, respectively (with a median of 3998 and 3338 genes per cell), performed principle component analysis on the scaled gene expression data, followed by UMAP visualization and unsupervised clustering. We assigned clusters to different cell types based on the co-expression of multiple marker genes (Fig. 9b, e; Supplementary Fig. 6a–c). In support of our immunostaining results, *Cep83* tKO embryos had an expanded Eomes/Tbr2-positive IP pool compared to control embryos, while the proportions of other cell types remained largely unaffected (Fig. 9c, d). Analysis of scRNAseq data comparing gyri to sulci in the gyrencephalic ferret model revealed an increased proportion in the expression of apical progenitor markers (Pax6) versus basal progenitor markers (Eomes/Tbr2) (Supplementary Fig. 6d). However, *Fgf10* tKO embryos did not show an expanded apical progenitor pool, presumably because the window of FGF10-mediated differentiation of apical progenitors was much earlier in development[24] (Supplementary Fig. 6e, f). *Cep83* tKO embryos also showed an increased proportion of upper layer neurons consistent with the expanded pool of IPs (Fig. 9e–g), an effect that was not seen in *Fgf10* tKO embryos (Supplementary Fig. 6g, h). Given that Flrt-expressing and non-expressing migrating neurons have different cell adhesion properties that influence their migration behavior[30], we quantified their proportions. We found that the numbers of Flrt3-mutant neurons (previously destined to express Flrt3), were increased compared to Flrt3-negative neurons in *Cep83* tKO embryos (60:40 proportion), and decreased in *Fgf10* tKO embryos (45:55 proportion) (Fig. 9h, i).

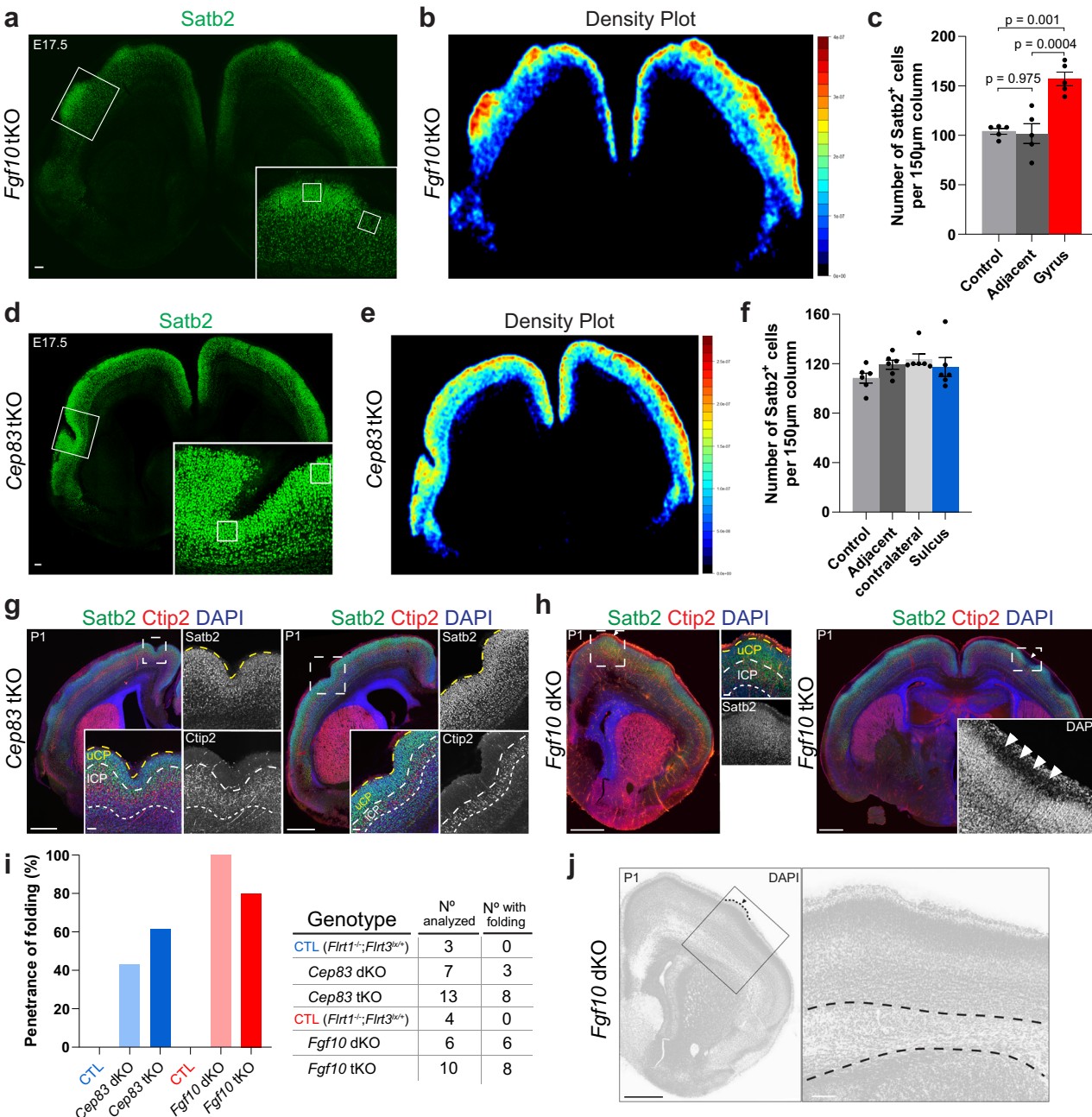

**Fig. 7 | Higher cell density in gyri of *Fgf10* tKO mice. a** Coronal brain sections of *Fgf10* tKO at E17.5, immunostained with Satb2. Gyrus in dashed rectangle is shown with higher magnification in the lower right corner. Small squares depict the region of interests (ROIs) used for quantification. Scale bar, 100 µm. **b** Heatmap of Satb2+ cell density (red, high, to blue, low) of the section shown in (**a**). **c** Quantification of the data shown in (**a**). Satb2+ cells in gyri, adjacent regions and in littermate controls (*n* = 5 embryos from 3 litter). Data are shown as mean ± SEM. one-way ANOVA with Tukey's post hoc analysis. **d** E17.5 *Cep83* tKO brain section with sulcus shown as in (**a**). **e** Heatmap of Satb2+ cell density of the section shown in (**d**). **f** Quantification of the data shown in (**d**). Satb2+ cells in sulci, adjacent and contralateral sides of cortex and littermate controls. (*n* = 6 embryos from 5 litters). Data are shown as mean ± SEM. No significant changes between groups. one-way ANOVA with Tukey's post hoc analysis. **g** Postnatal P1 *Cep83* tKO brain sections labeled with Satb2

(green), Ctip2 (red), and DAPI (blue). A sulcus in a dashed rectangle is shown with higher magnifications as Satb2 and Ctip2 single and merged channels; uCP and ICP are delineated by yellow and white dashed lines, respectively. Scale bars, 500 µm and 50 µm. **h** Postnatal P1 *Fgf10* dKO and *Fgf10* tKO brain sections labeled as in (**g**). Left: Gyrus-like protrusion in dashed rectangle is indicated with white arrowhead and shown with higher magnification as in (**g**). Right: section with a thicker marginal zone between two gyri, magnified to display the nuclei (arrowheads). Scale bars, 500 µm and 50 µm. **i** Folding penetrance of the indicated genotypes in postnatal P1 brains. CTLs are marked in the color of their respective mutants. Data from 3 litters of *Cep83* and 5 litters of *Fgf10* lines. **j** DAPI staining of postnatal P1 *Fgf10* dKO brain section in (**h**). The gyrus-like protrusion is magnified on the right and dashed lines indicate the corpus callosum. Scale bars, 100 µm.

For computational modeling, we also determined the overall numbers of DAPI+ cells in the cortex. The data revealed that the overall increases in cell numbers in the two tKO models varied considerably, with 9% in E13.5 *Cep83* tKO compared to 33% in E13.5 *Fgf10* tKO embryos (Fig. 10a). Similar to our previous modeling analysis[30], we

distributed particles representing Flrt1/3-positive (blue) and -negative cells (grey) in a 2D grid in a 50:50 ratio (Fig. 10b). The adhesive forces between cells were modeled as sine equations, reflecting the repeated patterns of cell clusters in Flrt3 gain- and loss-of-function experiments[30,41] (Fig. 10c, d). In the wild-type (balanced) situation, we

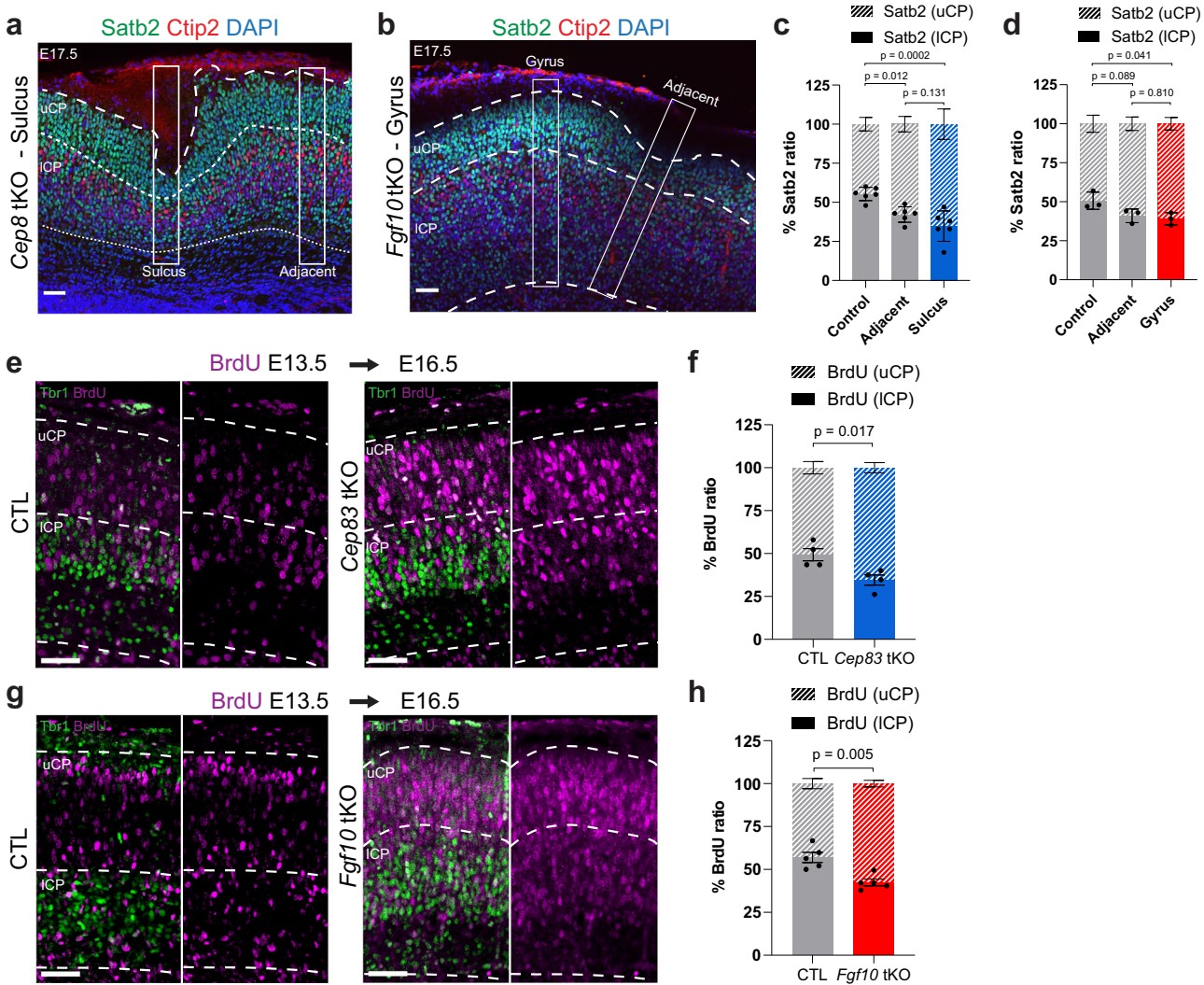

**Fig. 8 | Increased proportion of upper-layer neurons in the cortical plate of *Cep83* and *Fgf10* tKO embryos. a** Sulcus and adjacent region from an E17.5 *Cep83* tKO section and **b** Gyrus and adjacent region from an E17.5 *Fgf10* tKO section immunostained with Satb2 (green, upper CP) and Ctip2 (red, lower CP), DAPI (blue). White rectangles depict the ROIs used for quantifications shown in (**c, d**). Scale bars, 50 μm. **c, d** Quantification of data shown in (**a, b**). Proportion of Satb2+ cells in the upper and lower CP (defined by Ctip2 staining) in sulcus region, versus areas adjacent to the sulcus, and similar ROIs in CTL mice. Data are shown as mean ± SEM; Cep83 line: *n* = 6 embryos from 5 litters. Fgf10 line: *n* = 3 embryos from 3 litters. one-way ANOVA with Tukey's post hoc analysis. **e** BrdU injection was performed at E13.5 and analyzed at E16.5 in CTL and *Cep83* tKO embryos. BrdU expression was confirmed by immunostaining coronal sections with BrdU (magenta). The CP was subdivided into uCP and lCP using Tbr1 (green), and the number of BrdU+ neurons in each layer was quantified in (**f**). Scale bars, 50 μm. **f** Quantification of the data shown (**e**). *n* = 4 embryos from 3 litters. Data are shown as mean ± SEM. two-tailed t test with Welch's correction. **g** BrdU injection and analysis in CTL and *Fgf10* tKO embryos as described in (**e**). Scale bars, 50 μm. **h** Quantification of the data shown in (**g**). *n* = 5 sections of 4 embryos from 3 litters. Data are shown as mean ± SEM. two-tailed t test with Welch's correction.

determined that Flrt1/3-positive particles were attracted to one another (via Flrt-mediated homophilic adhesion) and the attraction force was higher than the one between Flrt1/3-negative cells (Fig. 10c). In the *Flrt1/3* dKO situation, we defined that Flrt1/3-mutant particles showed reduced attraction to one another (due to the loss of Flrt-mediated homophilic adhesion) (Fig. 10d). We had previously assumed that they were segregated due to repulsion from Flrt1/3-negative particles[30]. However, more recent analyses revealed that instead the segregation occurred because Flrt1/3-negative cells gained increased adhesion toward each other and that the adhesive force was stronger than homophilic adhesion between Flrt1/3 mutant cells (Y.-R.S., R.K., unpublished observations). These results suggest a model in which Flrt1/3 mutant cells are sorted by differential adhesion, consistent with the hypothesis of cell segregation by differences in intercellular adhesiveness[42].

We incorporated this new result into our model for *Flrt1/3* dKO and tKO mice, adjusting the amplitudes of the sine equations to increase the strength of the intercellular adhesion among Flrt1/3-negative cells in comparison to Flrt1/3 mutant cells (Fig. 10c, d). To analyze the behavior of the moving particles, they were set to move along the z axis, and both speed and attraction forces were random within a small range (ε) to resemble the fluctuations present in biological systems[43] (Fig. 10b). In the balanced situation, with a slightly higher attraction between Flrt1/3-positive particles compared to Flrt1/3-negative particles, the particles formed a rather smooth surface after moving along the z axis (Fig. 10e). In the *Flrt1/3* dKO situation with low attraction between Flrt1/3-mutant cells and strong attraction between Flrt1/3-negative cells, but no change in cell proportions or density, the surface became more wrinkled, consistent with the cortex folding phenotype (Fig. 10f, i). In the *Cep83* tKO situation with similar changes

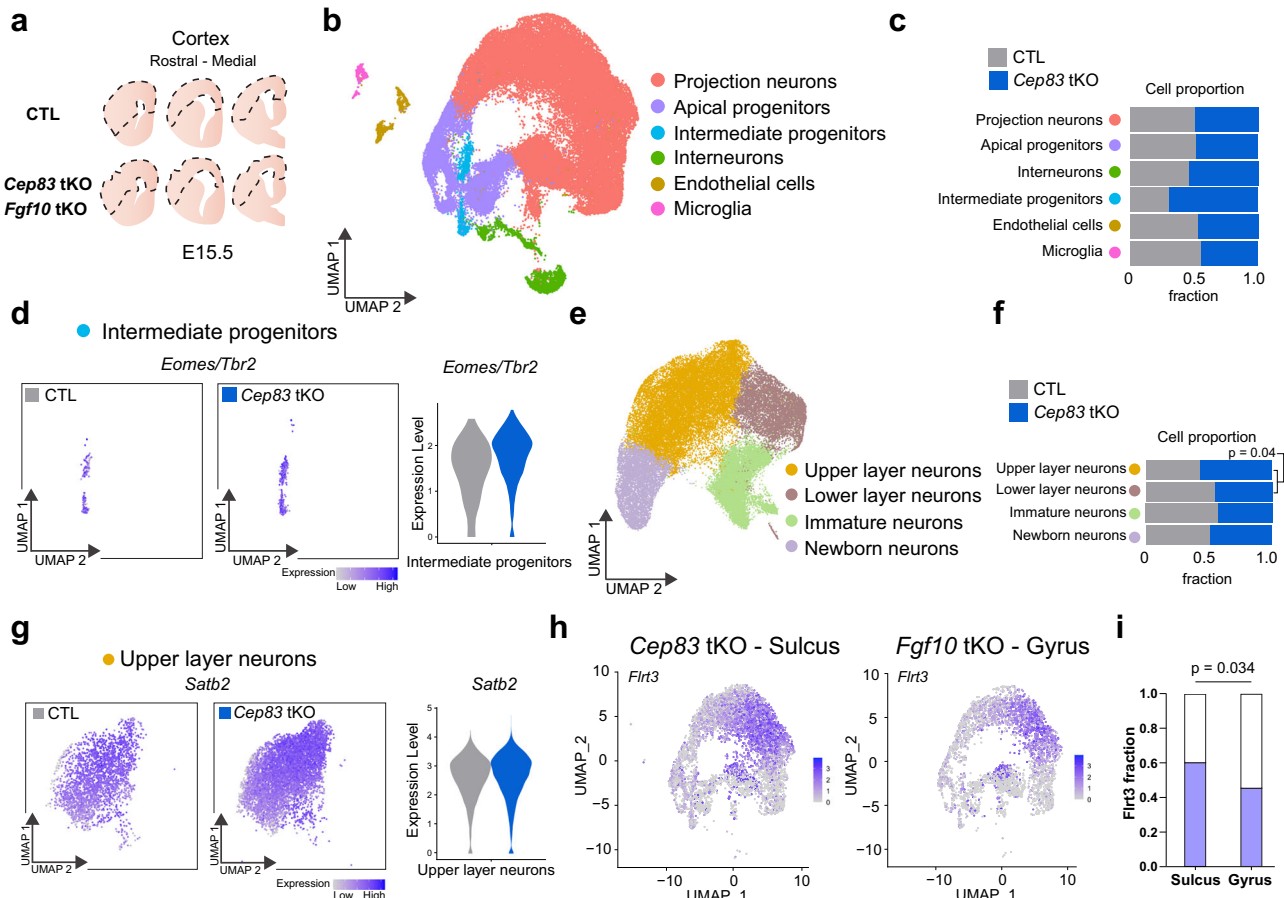

**Fig. 9 | scRNAseq analysis of *Cep83* and *Fgf10* tKO cortical populations in folded regions. a** Scheme of tissue collection for scRNAseq. Areas indicated by dashed line were manually dissected from coronal sections of E15.5 rostral to medial cortex. Three *Cep83* tKO brains (one with sulcus), three *Fgf10* tKO brains (one with gyrus), *n* = 2 littermate CTLs each were collected for tissue preparation. **b** UMAP clustering of cells from E15.5 *Cep83* tKO (*n* = 19267), *Fgf10* tKO (*n* = 14777), CTL (*n* = 22107) embryos classified as projection neurons (*n* = 40900), apical progenitors (*n* = 9212), interneurons (*n* = 2778), intermediate progenitors (*n* = 1474), endothelial cells (*n* = 1357), and microglia (*n* = 531). **c** Proportions of each cell type by genotype in CTL and *Cep83* tKO mice. **d** Expression of the IP marker gene Eomes (Tbr2) in intermediate progenitor cell type for CTL and *Cep83* tKO embryos. (UMAP and

violin plot). **e** UMAP clustering of projection neurons with annotated subpopulations, upper layer neurons (*n* = 17963), immature neurons (*n* = 10103), lower layer neurons (*n* = 6897), and newborn neurons (*n* = 5937). **f** Proportions of subpopulations by genotype in CTL and *Cep83* tKO embryos. Chi-Square test. **g** Expression of CP layer marker gene, Satb2, in upper layer neurons for CTL and *Cep83* tKO embryos. (UMAP and violin plots). **h** UMAP clustering for *Cep83* tKO embryo in sulcus area and for *Fgf10* tKO embryo in gyrus area. **i** Quantification of Flrt3+ cells in apical progenitors, intermediate progenitors, and projection neurons in the sulcus area of the *Cep83* tKO embryo and in the gyrus area of the *Fgf10* tKO embryo. Chi-Square test.

in attraction forces as in *Flrt1/3* dKO mice, a 60:40 proportion of Flrt1/3-mutant to negative particles, and an overall increase of 9% cell density, the surface became even more wrinkled, consistent with the enhanced cortex folding phenotype featuring mostly sulci (Fig. 10g, i). Conversely, the *Fgf10* tKO situation, with a 45:55 Flrt1/3-mutant to negative ratio and a 33% increase in cell density, resulted in a surface that had wrinkles, but also elevated areas, resembling the enhanced cortex folding phenotype featuring predominantly gyri (Fig. 10 h, i). When performing the simulations 25-times, gyrus-like structures were obtained in 76% of the cases in the condition representing the *Fgf10* tKO compared to 4-8% in other conditions (Fig. 10i). Taken together, the combination of modeling attraction forces, proportions of Flrt1/3-mutant to Flrt1/3-negative particles, and cell densities matched the experimental observations with dKO and tKO mice rather well by producing a more wrinkled surface with sulcus-like and gyrus-like features.

## Discussion
In the present study, the genetic combination of progenitor expansion with divergent migration allowed us to address a long-lasting question

concerning the mechanisms of cortex folding: what is the relative contribution of progenitor expansion and cortical migration to this complex process? Previously, we have shown that genetic deletion of Flrt1/3 adhesion molecules causes formation of cortical sulci resulting from increased lateral dispersion and faster neuron migration, in the absence of progenitor amplification[30]. Here, we find that combining the *Flrt1/3* dKO migration model with an additional genetic deletion that causes progenitor expansion in the same animal produces a synergistic effect that greatly enhances cortex folding. The main features of cortex folding, gyri versus sulci, correlate with the type of progenitor pool that was unevenly enlarged. Expansion of intermediate progenitors by deletion of Cep83 leads to an increase in Flrt-mutant neurons, thereby increasing cortex folding and the formation of sulci. Conversely, enlargement of the apical progenitor pool by deletion of Fgf10 results in fewer Flrt-mutant neurons, which enhances cortex folding by promoting gyrus formation. Cortex folding persists at least to postnatal P1 stage, demonstrating that these are not transient embryonic structures. Single cell transcriptomics and computational simulations identify key parameters that cooperate to promote cortical gyrification: changes in adhesive properties of cortical

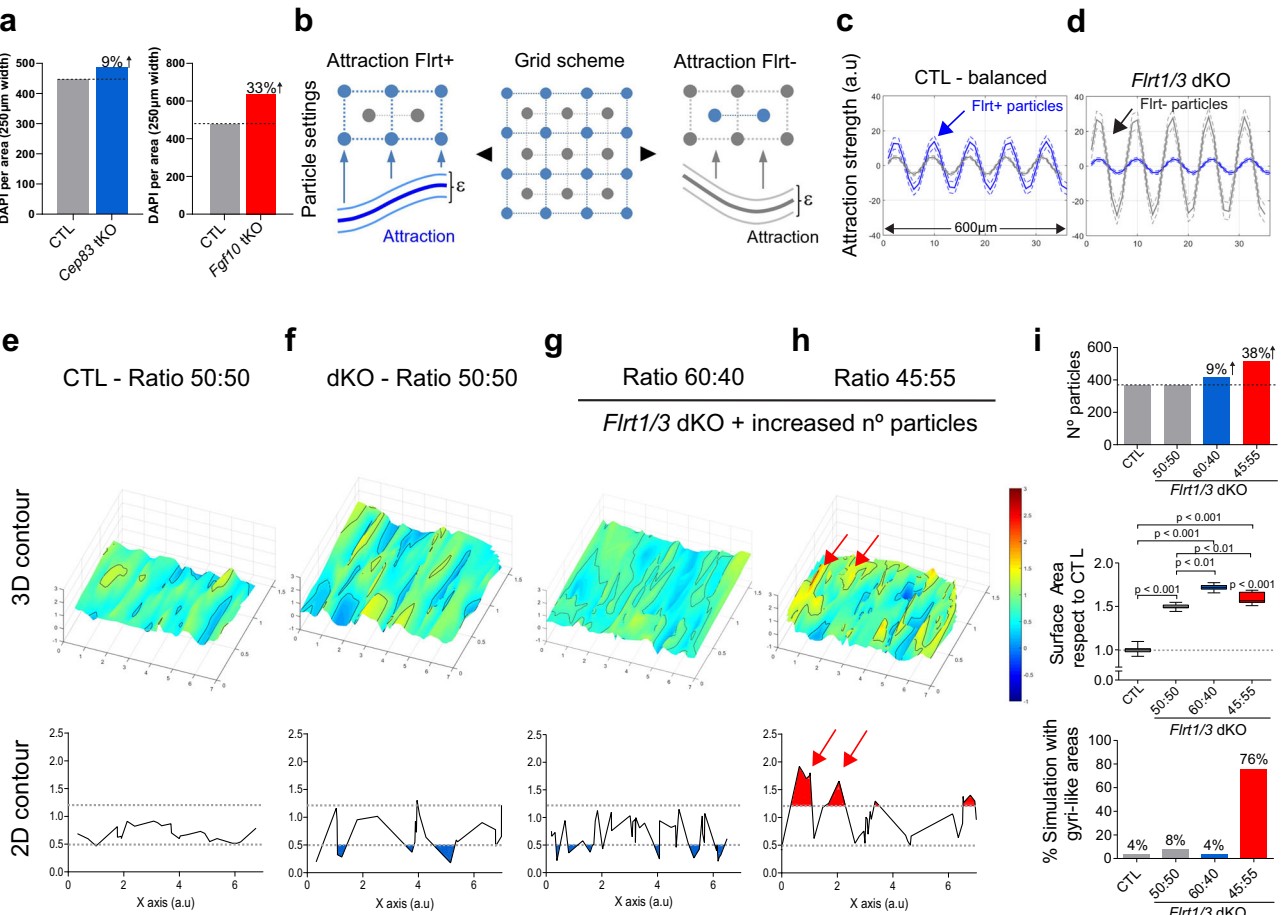

**Fig. 10 | Modeling cortex folding in *Cep83* tKO and *Fgf10* tKO mice. a** Number of cells based on DAPI staining per cortical area in CTL and *Cep83* tKO and *Fgf10* tKO mice. **b** Scheme illustrating the position of the Flrt-positive (blue) and –negative (grey) particles. Lines between the particles indicate attraction forces that are random within a small range (ε). **c, d** Sinus equations representing the attraction forces among Flrt-positive (blue) and negative (grey) particles in both control, (**c**) and *Flrt1/3* dKO condition, (**d**). In the latter, Flrt-negative particles show increased adhesion forces (higher amplitude) than Flrt-positive ones. Dashed lines indicate the small range of variation (ε). **e–h** Surface plots and line graphs of the distribution of the particles on the Z axis after computer simulations. The control situation with a 50:50 ratio of Flrt+ and Flrt-negative particles results in a uniform surface, (**e**). The

*Flrt1/3* dKO condition with altered attraction forces results in a wavy surface, (**f**). This effect is enhanced by increasing the number of particles and changing the proportions between Flrt-positive and -negative particles, **g, h** 2D plots representing the contour of a central slice through the surface plot are shown below. Deeper areas of the surface (below 0.5), mainly present in the (**f, g**) condition, are colored in blue. Elevated areas (above 1.2, representing gyri-like structures) are more prominent in the (**h**) condition; these are colored in red and marked with a red arrow. **i** Number of particles, surface area, and percentage of simulations with gyri-like areas (*n* = 25) calculated from conditions, (**e–h**). Data are represented as a box plot, with median (center line), interquartile range (box) and minimum and maximum values (whiskers). one-way ANOVA with Tukey's post hoc analysis.

---

neurons, their proportions and densities in the cortical plate, combined with lateral dispersion during their radial migration.

## Mechanisms underlying cortical gyrification

Most previous models of cortex folding involved the expansion of progenitors[9,10,17,44]. The *Flrt1/3* dKO mouse model was therefore quite unique in the sense that cortex folding was induced independent of progenitor amplification, by lateral dispersion and accelerated neuron migration[30]. This neuronal migration behavior remained an important feature of the novel tKO models used in the present study and apparently was affected neither by deleting Cep83 nor Fgf10. Likewise, the increased numbers of IPs for Cep83 and APs for Fgf10 reported in their single-KO lines[17,24] were similar to those observed in the tKO models, suggesting minimal effect by deleting Flrt1/3. Therefore, the combination of proliferation with divergent cell migration produces an enhanced cortex folding phenotype compared to the one produced by each of these two factors individually[17,24,30]. And yet, the folding phenotypes of the two tKO models were qualitatively very different. What could be the critical parameters determining the formation of sulci versus gyri?

In *Cep83* tKO embryos, where sulci are prominent, progenitor expansion starts around E12.5, involving IPs throughout the entire cortex. scRNAseq analysis revealed a higher proportion of Flrt3-mutant (previously destined to co-express Flrt1/3) to Flrt3-negative neurons (60:40) than in controls (50:50), accompanied by a modest 9% increase in cell density. Conversely, in *Fgf10* tKOs, which feature predominantly gyri, progenitor expansion happens early, involves NECs and apical progenitors, and is confined to the rostral parts of the cortex. scRNAseq analysis revealed a reduced proportion of Flrt3-mutant to Flrt3-negative neurons (45:55), coupled with a strong 33% increase in cell density.

Which of these changes may be most important? We have previously shown that sulcus formation in the *Flrt1/3* dKO model is highest in the lateral cortex where both proteins are strongly expressed[30,41]. In the *Cep83* tKO model, more Flrt-mutant cells are generated, and the phenotype extends into the lateral cortex, inducing cortical expansion in an area where divergent and faster neuronal migration is prominent due to the absence of Flrt1/3 proteins[30]. This may favor the strengthening of the phenotype already present in the *Flrt1/3* dKO, namely the formation of sulci. Conversely, in the *Fgf10* tKO model, fewer Flrt3-

mutant cells are generated compared to the *Flrt1/3* dKO model, weakening lateral dispersion and prompting neurons to taking a more radial migratory path. In addition, the effects of Fgf10 deletion are confined to the rostral cortex, where the expression of Flrt3 protein is low[30]. This local overproduction of predominantly radially (rather than tangentially) migrating neurons may favor the formation of gyri (see model in Supplementary Fig. 6i). Moreover, it is possible that differences in cell densities between the two models influence the formation of sulci versus gyri. Deletion of Cep83 resulted in a modest increase in cell density and the formation of sulci, whereas deletion of Fgf10 featured a large increase in cell density and the formation of gyri. This would be consistent with studies showing that in gyrencephalic species, regions in the cortical plate with lower and higher neuronal densities correspond to future sulci and gyri, respectively, during development[45].

Our model is also supported by computational simulations. Simply reducing intercellular adhesion between Flrt-mutant particles (as in the *Flrt1/3* dKO model) and increasing intercellular adhesion between Flrt-negative particles (Y.-R.S., R.K., unpublished observations), while keeping particle numbers unchanged, generated a more wrinkled surface. We propose a modification of our previous model[30] that Flrt1/3 mutant cells are sorted away from Flrt-negative cells by differential adhesion (rather than being subjected to increased repulsion from Flrt-negative cells) and that this effect favors lateral dispersion and faster migration. The folding (wrinkling) effect was enhanced when more Flrt-mutant particles were added, and the particle density was modestly increased as in the *Cep83* tKO model. In this case, the combination of more particles with reduced intercellular adhesiveness with a modest cell density increase favored sulci formation. The wrinkled surface contained elevated areas when fewer than normal Flrt-mutant particles were added, and the particle density was markedly increased as in the *Fgf10* tKO model. In that case, the combination of more particles with increased intercellular adhesiveness with a large increase in cell density, favored the formation of gyri.

Our findings provide a conceptual framework of how the gyrencephalic cortex structure is physically sculpted during development. It incorporates the roles of progenitor expansions and the effects these expansions have on overall cell density and on the proportions of cortical neurons with distinct adhesive and migratory properties. In the future, it would be interesting to test other combinations of genetic manipulations that expand different types of progenitors and lead to divergent neuron migration. This could include the use of other Cre lines, which may have different recombination onsets and thus target different progenitors. Additionally, the persistence of both sulci and gyri into postnatal stages provides the opportunity to confirm that the observed curvatures truly correspond to genuine and functional folds. Since the human gyrencephalic neocortex expresses much lower levels of Flrt1/3 compared to the lissencephalic mouse neocortex[30], these studies will also provide insights into the transition from gyrencephaly to lissencephaly during mammalian evolution.

## Methods

### Mouse lines
*Flrt1−/−; Flrt3lx/lacZ* mice[46,47] were crossed with either *Cep83lx/lx* mice[17] or *Fgf10lx/lx* mice[48], leading to *Cep83lx/lx;Flrt1−/−; Flrt3lx/lacZ* mice or *Fgf10lx/lx;Flrt1−/−; Flrt3lx/lacZ* mice. These animals were further crossed with either Emx1-Cre or Foxg1-Cre to remove the floxed alleles. Genotyping primers are available in Supplementary data sheet. All animal experiments were approved by the Government of Upper Bavaria and carried out in accordance with German guidelines for animal welfare. All mice (C57BL/6 and 129/SvJ mixed background) were housed with 12:12 h light/dark cycle and food/water available ad libitum in the facilities of the Max Planck Institute of Biological Intelligence. License number ROB 55.2-2532.Vet_02-20-2 until 16.10.2025. With regard to embryonic

stage, the midday of vaginal plug formation was regarded as embryonic day 0.5 (E0.5).

### Immunohistochemistry
For tissue preparation, pregnant mice were anesthetized using isoflurane and subsequently sacrificed by decapitation. Then, whole heads from E11.5, E13.5 mice and whole brains from E15,5, E17,5 mice were dissected and were fixed in 4% PFA in PBS at 4 °C overnight and stored in PBS. To obtain coronal brain sections, tissue was embedded in 4% agarose in PBS. Serial 100 μm (E15.5 and E17.5) and 80 μm (E11.5 and E13.5) thick sections were cut in PBS with a Leica VT1000 S vibratome. The sections were incubated with primary antibodies diluted with 0.2% BSA/0.5%Triton/5% donkey serum in PBS, including rabbit anti-Satb2 (1:500, Abcam), rat anti-Ctip2 (1:500, Abcam), goat anti-Sox2 (1:1000, R&D Systems), rat anti-Histone H3 (1:500, Abcam), rabbit anti-Tbr1 (1:500, Abcam), rabbit anti-Tbr2/Eomes (1:200, Abcam), rabbit anti-Cux1 (1:500, Santa Cruz), rabbit anti-Pax6 (1/300, BioLegend), rat anti-BrdU (1:100, Abcam), rabbit anti-cleaved Caspase3 (1:300, Cell Signaling Technology, Inc), mouse anti-Pvim (1:500, Abcam) and rabbit anti-BLBP (1:300, Millipore). Secondary antibodies were diluted at 1:500 with 0.2% BSA/0.5%Triton/5% donkey serum in PBS and incubated for 2 hr at room temperature. The secondary antibodies used were Cy3-conjugated donkey anti-rat IgG, Cy3-conjugated donkey anti-rabbit IgG, Cy3-conjugated donkey anti-mouse IgG, Alexa 488-conjugated donkey anti-rabbit IgG, Alexa 488-conjugated donkey anti-rat IgG, and Alexa 674-conjugated donkey anti-goat IgG (Jackson Immunoresearch Laboratories). Cell nuclei were counterstained with 4′,6-Diamidino-2-phenylindole dihydrochloride (DAPI)/PBS (1:1000; Invitrogen). For observation, sections were mounted with DAKO mounting medium and sealed with cover slides by Menzel-Gläser (Menzel GmbH & CO). Images were visualized using a Leica SP8 confocal microscope.

### In utero electroporation assays
In utero electroporation was performed as described previously with minor modifications[49]. Timed pregnant mutant mice at E13.5 were anesthetized with isoflurane (CP-pharma) using a Fluovac 34-0387 (Harvard Apparatus) and Vevo Compact Anesthesia System (VisualSonic). For surgery, the uterus of the mouse was exposed after making an approximately 3 cm incision in the middle abdominal region. Then, both sides of uterine horns were pulled out, and ~1–2 μl of a mixture containing 1 μg/μl pCAG-Ires-GFP plasmid (Addgene, Cat #11159) and 1% Fast Green (Sigma, final concentration 0.2%) in PBS was injected into the lateral ventricle of embryos with a mouth-controlled glass capillary pipette. Immediately, square pulses (30 V, 50 ms, six times at 1 s intervals) were delivered into embryos with an electroporator (ECM 830, BTX) and a forceps-type electrode (CUY650P5, NepaGene). After the electroporation, the uterus was returned back inside the abdomen using ring forceps and the incision was sutured with PERMA-HAND Seide (9.3 mm diameter curved needle, 45 cm of thread, Ethicon). After surgery, mice were placed on a 37 °C heating pad for recovery and kept until the desired embryonic stage.

### BrdU analysis
A single injection of 5-bromo-20-deoxyuridine (BrdU, Sigma-Aldrich) dissolved in PBS, 0.15–0.2 ml at a concentration of 10 mg/ml BrdU was administrated to pregnant females at E11.5 and E13.5, intraperitoneally for a short BrdU pulse analysis, to give a final concentration of 50 mg per g of mouse weight. Pregnant females were then anesthetized using isoflurane and subsequently sacrificed by decapitation after 30 min or 3 days later. Brains were collected and fixed overnight with 4% PFA and stored in PBS. For staining with the BrdU antibody, sections were pretreated with 2 N HCl for 30 min at RT and washed with Na2B4O7 (pH 8.5) twice for 15 min.

## Time-lapse experiments

Mice were anesthetized using isoflurane and subsequently sacrificed by decapitation. Embryo cortices were electroporated at E13.5 with pCAG-Ires-GFP (Addgene Cat #11159). After 2 days, E15.5 of embryonic brains were isolated in an ice cold sterile filtered, and aerated (95% O2/ 5% CO2) dissection medium (15.6 g/l DMEM/F12 (Sigma), 1.2 g/l NaHCO3, 2.9 g/l glucose (Sigma), 1% (v/v) penicillin streptomycin (GIBCO)). Using 4% low melting agarose (Biozym), brains were embedded for cutting into 300 mm thick sections using a vibratome (Leica, VT1200S). Sections were immersed in a collagen mix (64% (v/v) cell matrix type I-A (Nitta Gelatin), 24% (v/v) 5x DMEM/F12, 12% (v/v) reconstitution buffer (200 mM HEPES, 50 mM NaOH, 260 mM NaHCO3) and transferred onto a cell culture insert (Millicell, PIC-MORG50). Sections were then incubated at 37 °C for 10 min to solidify the collagen mixture. 1.5 mL slice medium (88% (v/v) dissection medium, 5% (v/v) horse serum, 5% (v/v) fetal calf serum, 2% (v/v) B27 supplement (GIBCO), 1% (v/v) N2 supplement (GIBCO)) was added to the dish surrounding the culture insert and incubated for minimum 30 min at 37 °C. Before imaging, culture medium was added to the top of the sections to allow objective immersion. Imaging was performed using a 20x water immersion objective on a Leica SP8 confocal microscope system equipped with a temperature-controlled carbon dioxide incubation chamber set to 37 °C, 95% humidity, and 5% CO2. Sequential images were acquired every 15 min for 24–48 h. After imaging, slices were genotyped to identify triple knock outs and littermate controls. The neuronal movement was tracked using the software Fiji with plugin 'Manual Tracking.' Only neurons entering the CP were tracked. Tracking was stopped when cells reached the upper cortical plate Single cell track analysis and plotting was carried out using custom made Python scripts.

## 3DiSCO tissue clearing

A modified clearing protocol was used based on Belle et al., 2017[50]. Mice were anesthetized using isoflurane and subsequently sacrificed by decapitation. Briefly, E17.5 embryo brains were fixed in 4% PFA overnight at 4 °C and stored in PBS. First, whole brain immunostaining was performed before tissue clearing. Brains were incubated in blocking buffer for 24 h at RT on a horizontal shaker. Then brains were stained with primary antibodies, rabbit anti-Satb2 (1:1000, Abcam) and rat anti-Ctip2 (1:1000, Abcam) over 7 days with blocking buffer containing saponin at 37 °C on a shaker. Next, brains were washed for 1 hour 6 times in 15 ml falcon tubes containing blocking buffer. Then, brains were incubated at 37 °C on the horizontal shaker over 2 days with secondary antibodies Alexa Fluor 647 and 594 (1:500, Jackson Immunoresearch Laboratories), which had been filtered with a 0.20 mm filter and diluted in blocking buffer with saponin 0.1%. The brains were washed for 1 h 6 times in 15 ml Falcon tubes filled with blocking buffer. Then, tissue clearing was performed. Briefly, the whole stained brains were immersed in 50% tetrahydrofurane (THF, Carl Roth) overnight, in 80% THF for 1 h, in 100% THF for 1 h, then in fresh 100% THF for another hr, then in 100% dichloromethane (DCM, Carl Roth) until the brains sank, and finally in 100% dibenzylether (DBE, Sigma), followed by another step of fresh 100% DBE. Then, cleared brains were imaged with a 4× objective lens (Olympus XLFLUOR 340) equipped with an immersion-corrected dipping cap mounted on a LaVision UltraII microscope coupled to a white-light laser module (NKT SuperK Extreme EXW-12). Images were taken with 16-bit depth and at a nominal resolution of 1.625 μm per voxel on the x and y axes. Brain structures were visualized with Alexa Fluor 594 (using a 580/ 25 nm excitation filter and a 625/30 nm emission filter) and Alexa Fluor 647 fluorescent dye (using a 640/40 nm excitation filter and a 690/ 50 nm emission filter) in sequential order. Laser power was set for each channel so as not to exceed the dynamic range of the Neo 5.5 sCMOS camera (Andor). For 12× imaging, we used a LaVision objective (12×/ 0.53 NA MI PLAN with an immersion-corrected dipping cap). Camera

exposure time was set to 105 ms and 90 ms for the 4× and 12× imaging, respectively. In the z dimension, images were taken in 5 μm and 2 μm steps, while using left- and right-sided illumination for the 4× and 12× imaging, respectively. Our nominal resolution was 1.625 μm × 1.625 μ m × 5 μm and 0.602 μm × 1.602 μm × 2 μm for the x, y and z axes, with 4× and 12× objectives. The thinnest point of the light sheet was 28 μm and 9 μm.

## Fluorescence in situ hybridization

Mice were anesthetized using isoflurane and subsequently sacrificed by decapitation. Embryonic brains (E11.5 and E15.5) were fixed in 4% PFA in PBS at 4 °C overnight and placed in 30% sucrose/PBS (weight/ volume) until the tissue sank (12 h-16hrs). After cryopreservation, the brains were embedded in O.C.T. Compound (Sakura Finetek), frozen on dry ice, and stored at −80 °C. Coronal brain sections (14 μm) were cut using a Leica CM3050S cryostat, mounted onto Superfrost Ultra Plus slides (Thermo Scientific), and stored at -20 °C. Frozen sections were air-dried for approximately 30 min. For FISH analysis, RNAscope Fluorescent Multiplex Assays (ACD, 320850, 322000, and 322340) were conducted according to the manufacturer's instructions (ACD, 320293-USM and 320535-TN) with RNAscope Probes directed against Fgf10 (ACD, 446371), Flrt1 (ACD, 555481), and Flrt3 (ACD, 490301). Immunostaining against CEP83 protein was performed prior to counterstaining with DAPI for *Cep83* tKO embryos.

## Single-cell RNA sequencing and sample preparation

Mice were anesthetized using isoflurane and subsequently sacrificed by decapitation. E15.5 of mutant embryos were removed from the uterus of mutant females, and stored in ice-cold in Leibowitz medium with 5% FBS. Brains were isolated and cut in 300 μm using a vibratome (Leica VT1000S, Germany) in cutting buffer, Leibowitz medium with 5% FBS. Then, brain sections were manually dissected for both hemisphere of cortical primordium of cortex. Meanwhile, genotyping was performed. Then, collected tissues were transferred into a dissociation Buffer, EBSS. Dissociation was performed with Worthington Kit and manually Papain dissociation system was carried out according to the recommended protocol (Worthington, #LK003163).

## Single-cell RNA library preparation

For experiments utilizing the 10x Genomics platform, the following reagents were used Chromium Single Cell Next GEM Single Cell 3' GEM kit v3.1 (PN-1000130), Library Construction Kit (PN-1000196), Chromium Next GEM Single Cell 3' Gel Bead Kit v3.1 (PN-1000129) and Single Index Kit T Set A (PN-1000213) and Dual Index Kit TT Set A (PN-1000215) were used according to the manufacturer's instructions in the Chromium Single Cell 3' Reagents Kits v3.1 User Guide.

## Sequencing and data analysis

Transcriptome and barcode libraries were sequenced either on an Illumina NextSeq 500 and NovaSeq 6000 at the Next Generation Sequencing Facility of the Max Planck Institute of Biochemistry. Then, the sequencing data were processed with cellranger (version 7.0.1, reference 'refdata-gex-mm10-2020'). Further processing was performed in R using Seurat (version 4.2.0) as follows: first, we merged the cellranger output files. We kept cells with more than 200 RNA molecules, less than 20% mitochondrial, more than 5% ribosomal, and less than 20% hemoglobin content. We used cellranger (version 7.0.1) to extract fastq files, align the reads to the mouse genome (10x genomics reference build MM10 2020 A), and obtain per-gene read counts. Subsequent data processing was performed in R using Seurat (version 4.2.0) with default parameters if not indicated otherwise. After merging the data, we normalized the data (normalization.method = 'Log-Normalize', scale.factor = 10000), detected variable features (selection.method = 'vst', nfeatures = 2000), and scaled the data (vars.to.regress = c('nCount_RNA')). We then applied quality control

filters on cells with the following criteria: a) more than 200 genes detected b) less than 20% mitochondrial genes reads c) more than 5% ribosomal protein genes reads d) less than 20% hemoglobin genes reads e) singlets as determined by doubletFinder (version 2.0.3, pK = 0.09, PCs = 1:10). After performing principal component analysis on variable features, nearest-neighbor graph construction and UMAP dimension reduction were carried out on PCs 1-20, followed by cell clustering at a resolution of 0.2. Neuronal subset clustering was performed with are resolution of 0.5. Single-cell RNA-seq data for gyri and sulci from ferret samples were obtained from the published NCBI Gene Expression Omnibus with accession numbers: GSE234305[51]. We used the same metadata and cluster categorization provided by the authors to quantify the expression of the apical (Pax6) and basal (Eomes/Tbr2) gene markers in the germinal zone (average expression from VZ + SVZ) of both gyri and sulci. We then calculated the expression proportion between these two markers for both regions.

### Computational simulations

We modelled the attraction forces for Flrt-positive, -mutant and -negative cells as sine curves based on in vivo gain- and loss-of-expression experiments[30,41], using previous code (https://doi.org/10.5281/zenodo.15583689). The equation had the following terms: Attraction curve = κ[Asin(λx)]+bs+ε, where A is the amplitude and λ the frequency. The frequency value was the same for both attraction curves (λ, 0.051) but not their amplitude (A, Flrt-positive and mutant: 41,41, Flrt-negative:56.74). The strength of both curves was adjusted with the term κ to represent both: the balanced situation with slightly higher attraction for Flrt-positive particles (k, Flrt-positive: 1/4, Flrt-negative: 1/10), and the *Flrt1/3* dKO situation with low attraction between Flrt-mutant particles and strong attraction between Flrt-negative particles (k, Flrt-positive: 1/10, Flrt-negative: 1/2). The basal subtraction value (bs, 100) was used for fitting both curves; factor ε changes randomly from −10 to 10 and is used to add noise.

We used particles to represent Flrt-positive, -mutant, and Flrt-negative cells using the particle system toolbox, MATLAB particles version 2.1[52]. Flrt-positive or -mutant particles were arranged in an 8 by 36 matrix, spaced by 0.2 units[30], and were given an attraction towards neighboring particles in both axes using their attraction curve. Flrt-negative particles were arranged in a matrix shifted 0.1 units on both the X and Y axes relative to the previous matrix (50:50 condition) and were attracted to neighboring particles in both axes using their attraction curve. The number of particles in both matrices was adjusted independently to mimic the overall increase in particles and the Flrt-mutant to Flrt-negative ratio observed in experiments with tKO mice.

All particles received random speeds (ranging from 6 to 12 arbitrary units) for moving along the Z axis and were simulated for 100 frames (0.001 units step time). After simulation, the position of every particle was retrieved, and their area was calculated by triangulating the surface formed by their positions and then summing the areas of the triangles. The 3D surface and 2D contour lines representing the central slice of the surface plot at the median Y value were plotted in MATLAB.

### Quantification and statistical analysis

Images were processed with the open-source image analysis software Fiji. Automatic cell counting analysis was performed using open-source CellProfiler 4.2.5 and software Fiji with plugin 'Cell counter'. Kernel Density Estimation (Heat map) was performed using R Studio. Statistical significance was determined using one-way ANOVA Tukey's post hoc test or *t*-test with Welch correction or Chi-squared test, or Wilcoxon signed rank test with continuity correction using Prism version 9,10 (Graphpad Software) or R studio. Statistical significance was defined as $p < 0.05$. All values in the text and in the figure legends indicate mean +/- SEM.

### Reporting summary

Further information on research design is available in the Nature Portfolio Reporting Summary linked to this article.

## Data availability

The scRNA datasets used in this study are publicly available at the NCBI Gene Expression Omnibus (GEO) database (www.ncbi.nlm.nih.gov/geo) with accession numbers GSE267391. Source data are provided with this paper.

## Code availability

The Time-lapse experiments, Single cell track analysis data generated in this study are provided in this study are available https://doi.org/10.5281/zenodo.15522270. The MATLAB base code for Computational simulations data generated in this study is available https://doi.org/10.5281/zenodo.15583689.

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

## Acknowledgements

We thank S. Cappello for her insightful comments during the course of this study, C. Mayer, A. Bright, Y. Kotlyarenko, P.A. Morales, and H. Lim for help with scRNA sequencing experiment, K. Voelkl and L. Schaffmayer for technical help with scRNA sequencing experiment. D. Feigenbutz for help with data processing of scRNA library. This study was supported by the Max-Planck Society, by the Ministry of Science and Technology of China (2021ZD0202300) and New Cornerstone Investigator Program (S.-H.S.). by the Deutsche Forschungsgemeinschaft (DFG, German Research Foundation) under Germany's Excellence Strategy within the framework of the Munich Cluster for Systems Neurology (EXC 2145 SyNergy – ID 390857198).

## Author contributions

S.H.C., S.-H.S., and R.K conceptualized the study and designed experiments; S.H.C. performed most of the experiments; D.E.Y. helped with phenotypic analysis of embryonic and postnatal mutant mice. D.S.D.A helped with phenotypic analysis of some of the mutant mice and time-lapse experiment; M.I.T. and A.E. performed 3D imaging; T.S. performed scRNAseq data analysis; T.R. performed time-lapse speed analysis; W.S. and J.Y. helped with the analysis of *Cep83* cKO mice. G.S.B. helped with ISH and immunostainings; Y.-R.S. assisted in scRNAseq experiment; D.d.T. performed computational simulations; R.K. and D.d.T. supervised; R.K., S.H.C., and D.d.T. wrote the manuscript with help from all authors; R.K. provided funding.

## Funding

## Competing interests

The authors declare no competing interests.
