## [Transparent Peer Review file · Nature Communications]

Cortex Folding by Combined Progenitor Expansion and Adhesion-Controlled Neuronal Migration

Corresponding Author: Professor Rüdiger Klein

Version 0:

Reviewer comments:

Reviewer #1

(Remarks to the Author)

Chun et al. investigated the mechanisms underlying cortical folding using various kinds of mice. They examined the phenotypes of *Cep83/Flrt1/3-tKO* and *FGF10/Flrt1/3-tKO* mice and found that these mice exhibited cortical fold-like structures. They proposed that expansion of IP cells leads to sulcus formation, whereas expansion of apical progenitors results in gyrus formation. Although their findings are potentially interesting, and their data sets seem to be based on a huge amount of experiments, their findings unfortunately did not seem sufficient for significantly deepening our understanding of the mechanisms underlying cortical folding.

Major points

1. The authors nicely showed that *Cep83/Flrt1/3-tKO* and *FGF10/Flrt1/3-tKO* mice exhibit cortical fold-like structures. However, previous reports have already shown that *Cep83-KO*, *FGF10-KO* and *Flrt1/3-KO* mice also have fold-like structures. Although the phenotypes of *Cep83/Flrt1/3-tKO* and *FGF10/Flrt1/3-tKO* mice were much stronger than *Cep83-KO*, *FGF10-KO* and *Flrt1/3-KO* mice, the authors' findings do not seem to lead to a significant conceptual advance of our understanding of the mechanisms underlying cortical folding.
2. In Figures 2 and 4, the authors counted progenitor cells using *dKO* and *tKO* mice. The results seemed similar to those previously obtained using *Cep83-KO* and *Fgf10-KO* mice, and it was unclear what new information these results add compared to the previous results. It would be intriguing to examine the synergistic effects of *Cep83-KO* and *Flrt1/3-KO* on progenitor numbers, and those of *Fgf10-KO* and *Flrt1/3-KO* on progenitor numbers. In addition, because the authors' data showed that cortical folding was more prominent in the rostral cerebrum than in the caudal cerebrum, it would be intriguing to compare the increase in progenitor cells between rostral and caudal forebrains and examine the correlation between the degree of cortical folding and the degree of increase in progenitor cells.
3. Physiological folds exhibit curvature not only at the cortical surface but also at the border between gray and white matter. The authors should examine if the border between gray and white matter also shows curvature in *dKO* and *tKO* mice. The pictures in Figures 3K, 3L and 5B showed the curvature at the surface but not deep in the cortex. Figures 3K, 3L and 5B seemed to show heterotopia rather than folds.
4. The authors proposed that expansion of IP cells leads to sulcus formation, whereas expansion of apical progenitors results in gyrus formation. Is this consistent with the data from gyrencephalic animals? The authors should show that IP cells and apical progenitors are abundant in sulcal regions and gyral regions, respectively, in developing gyrencephalic brains.
5. In Figure 6F, the authors showed that *Cep83-tKO* mice had increased upper layer neurons, while the authors concluded that *Cep83-tKO* leads to sulcus formation. Are upper layer neurons increased at sulci in gyrencephalic animals?
6. In Figure 6L-R, the authors tried to provide a theoretical basis for distinguishing gyrus and sulcus formation. I understood that these parameters affect the morphology of folds, but it was difficult to distinguish gyri and sulci from their modeling. The authors should provide more convincing modeling data sets.

Minor points

1. The authors showed the penetrance and the GI in Figures 1 and 3. In addition to this, it would be intriguing to examine the number of folds per section per each animal.
2. Fig 1E seemed incomplete. Some bars (e.g. median, whisker) seem to be missing.
3. Although the authors wrote that cortical layers were preserved (page 5, third line from the bottom; page 7, second line from the bottom), it was difficult to distinguish cortical layers in the pictures. The authors should provide more convincing pictures showing all cortical layers and the border between gray matter and white matter. Separate immunostaining images rather than merged images would be better.
4. The names of genes should be consistent. Different formats were used: "TBR2/Tbr2," "pH3/PH3," "pVIM/Pvim," "Sox2/SOX2," "Satb2/SATB2," "FLRT/Flrt" etc.
5. In the text and in Figures 2E and 2F and Extended data Figure 4E, the authors counted bRG cells by examining the number of pH3+/pVim+ cells. However, pH3+/pVim+ cells represent a subset of bRG cells. The authors should use appropriate markers for counting bRG-like cells.
6. The authors drew dashed lines to indicate the borders between cortical layers, but the reasons why they were able to draw the lines at specific positions were often unclear. For example, in Figure 5H, the border between the uCP and the iCP and that between the iCP and white matter seemed unclear. The authors should provide clearer evidence.
7. The authors should confirm that apoptosis did not occur in sulcal regions in Cep83/Flrt1/3-tKO and FGF10/Flrt1/3-tKO mice.
8. It seemed that "SATB+" should be "SATB2+" in Figure 5C and 5F.
9. Reference #52 did not have enough information.

Reviewer #2

(Remarks to the Author)

This is a fascinating study from the Klein lab, building on their previous findings about mechanisms of cerebral cortex folding. They previously showed that specific mechanisms and patterns of neuron migration are sufficient to drive folding of the mouse cortex, and here they show that combining the effect of neuron migration with increased cell proliferation exacerbates the cortex folding phenotype. The authors combine their previous mutant mouse model (Flrt1/3 dKO) with two additional conditional mutant mouse lines where progenitor proliferation is enhanced (Cep83 KO, Fgf10 KO). In both triple mutants, they show very robust and abundant folding, much more than in the separate mutant conditions. Intriguingly, however, in one case (Cep83) they show the seeming formation of sulci, whereas in the other (Fgf10) they show the formation of gyri. This is fascinating because then they show that the immediate effect of these mutations is different in each case, thus suggesting the existence of distinct and complementary mechanisms for the formation of folds and fissures in gyrencephalic mammals. The study ends with an analysis of cell types with altered abundance in each mutant condition, and a mathematical model aimed at supporting a hypothesis on how the multiple changes found in these mutants may collectively explain and contribute to the development of cortical folds. In my opinion, this is a perfect fit for Nature Communications and I am convinced that it will become a reference paper in the field. I would only recommend to tackle a handful of formal aspects, as detailed next.

The authors conclude their Intro by stating "Our results identify key developmental mechanisms that collaboratively drive cortex folding". This should be adjusted, as proliferation and cell migration had already been experimentally demonstrated as mechanisms that drive cortex folding, and their interaction/synergy in this process had been proposed. The novelty of this study is to demonstrate that the combination of these mechanisms enhances their effect on folding beyond the individual effects, and that the formation of gyri may involve slightly but significantly different mechanisms from the formation of sulci.

In the first paragraph of page 6, the authors indicate that Cep83 tKO leads to a strong cortex folding phenotype with the presence of bilateral sulci. However, it is not shown nor demonstrated that any sulci are bilateral in these mutants. The observations should be referred to as "folding phenotype in both cerebral hemispheres".

Last paragraph of page 6: "we estimated the number of PH3+ basal cells stained for phospho-vimentin (pVIM), which is known to be expressed in RG cells during cell division". While this is true, pVim stain is not specific to aRG but to all mitotic progenitor cells in the cerebral cortex, namely including IPCs. The standard criteria to identify bG are positivity for PH3 (or pVIM) and Pax6 outside of VZ. Thus, these observations must be rephrased and the conclusions re-considered.

"Collectively, these results indicate that deletion of Cep83 causes an expansion of IPs and bRG cells at E13.5" - As mentioned in the previous point, to support this conclusion the authors must show increases in basal PH3 that are positive for Tbr2 (IPC) or Pax6 (bRG). Unfortunately, their current results only show increased Tbr2+ cells (not Pax6!), and then increased basal proliferation but with no cell type-specific markers. pVIM labels any cortical mitotic cell, as shown by multiple authors previously, while bRGs are identified by the presence of a basal pVIM+ process.

In Fig 2, the authors indicate that quantifications were done on the most lateral cortex (almost ventral), whereas the folding

phenotype of these mutants is most obvious in parietal and dorsal regions. These quantifications should be re-done in the proper areas to proof the point that such changes in proliferation are related to such gyrus formation. Second, given the concepts of patterned proliferation and gene expression correlated with folding, mentioned in the Intro, does proliferation play a specific pattern along the latero-medial axis of the embryonic cortex of these mutants, which would explain the formation of these sulci?

Why Fgf10 Ix/Ix mice had 10% folding? Aren't these control embryos?

Panoramic views in Fig 3J fail to reveal folds or fissures; should be replaced by examples where this is somewhat visible.

Thickness of Satb2 layer is clearly larger in mutants than controls. Does this also affect the Ctip2 layer? This may be expected given the early recombination by FoxG1-Cre.

The authors report an expansion of Sox2+ apical progenitors in Fgf10 dKO and tKO mice compared to littermate controls (Flrt1^{-/-};Flrt3Ix/+). The views displayed in Fig. 4A clearly show that this analysis was done in the most rostral part of the telencephalon (no basal ganglia nor thalamus visible). Analysis of this region in the coronal plane is bound to high variability, because sections are far from perpendicular to the tissue and thus cortical thickness (and cell number) varies much from one section level to the next. As shown in Figure 1, the cortex folding phenotype extends from rostral to caudal cortex in these mutants, so the authors can choose where is most reliable to do the analysis. For these reasons, quantifications should be done in a more intermediate level of the telencephalon.

The results presented in Figure 5 are quite confusing. First, Ctrl embryos have very few apical mitoses, which concomitantly leads to a 50-50% proportion apical-to-basal. This is very much unlike previous observations from many different labs at this same developmental age, and should be revised. Second, dKO and tKO have a clear increase in Tbr2+ cells as shown in the example images, but this is not reflected in the quantification of Fig 4B; in addition, dKOs and tKOs virtually lack basal mitoses, which is particularly dramatic in tKOs. As mentioned in a previous comment, the authors need to clarify what are these Tbr2+ cells really: newborn neurons? IPCs stalled somewhere in the cell cycle? If ablation of Fgf10 expands apical progenitors, why are apical mitoses not significantly increased in tKOs?

In page 10: "Cep83 tKO embryos also showed an increased proportion of upper layer neurons" - The difference is very mild, and to draw such conclusion one should test statistical difference.

Page 13: "This would be consistent with studies showing that in gyrencephalic species, regions in the cortical plate with lower and higher neuronal densities correspond to future sulci and gyri, respectively, during development". - Directly related to these observations are some questions very relevant to understand the underlying mechanisms: are there differences in proliferation, IPC or bRG abundance along the developing cortex of these mutant mice? If yes, do these spatial differences explain the patterns of gyri/sulci? Are the trajectories of radial glia fibers modified as to explain the formation and location of gyri/sulci? The authors should address these questions, or at least discuss them with due consideration.

Reviewer #3

(Remarks to the Author)

In this manuscript the authors cross several mouse lines known to elicit a phenotype in in cortex folding to examine possible additive or synergistic effects. The rationale is that deletion of Flrts affects mostly migration and adhesion/repulsion, while the Cep83 or FGF10 deletion affects stem progenitor cells, such that affection both may cause more severe effects. Indeed, this is the case in triple mutants for Cep83 and Flrt1/3 with impressive folding (mostly sulci formation) in the triple mutants. The authors further demonstrate an accompanying increase in basal mitoses and Tbr2+ cells, i.e. basal progenitors, but not apical stem/progenitor cells. If this is causative for the more severe folding phenotype has not been determined. The second set of data crossing the Flrt1/3dKOs with FGF10cKO is less convincing, in particular in regards to what is referred to as "a fold" (see below). This is the weak point, as it casts doubts on the interesting conclusion that Cep83 deletion in combination with Flrt1/3 affects selectively sulci versus the combination with FGF10 deletion affecting gyri formation. The phenotypes are clearly different, but if the condensates of cells observed in the FGF10 deletion together with Flrt1/3 are really gyri is highly doubtful. Moreover, the authors focus on a single time point, collecting embryos just prior to birth, but it would be very important to know about later phenotypes – are they persisting or transient in nature.

Taken together, this is a highly promising manuscript that could become a broadly important piece of work if the questions below can be answered.

Suggestions:

- 1) For all histograms please show the data points in the histogram and provide information about the number of embryos analyzed from how many different litters (e.g. Figure 2, 4, 5, Figure S5).
- 2) In Figure 3C the authors outline a fold with a dashed line. However, there are clearly Satb2+ cells on top of the line in the lower left corner, raising doubts about this classification.
- 3) In Figure 3J the macroscopic sulci and gyri are not visible. Please provide better examples, if they exist.
- 4) Also in Figure 4K there is largely a dark space in SATB2-staining between the alleged "fold", but these may be weaker labelled SATB2+ cells (as suspected from the example in 3C), or if not, which cells are there? Other neurons/no neurons? There is a huge black space between CTIP2+ band and the pial surface that is very smooth – what is in this black hole?

5) It is understandable, but also problematic that the authors use different Cre driver lines for the Cep83 versus FGF10 deletions. Understandable, because they use the lines that have been used before in the single cKOs, but problematic, because we formally do not know if the differences in phenotype may relate to a different onset in the timing of recombination. I therefore suggest to do some pilot crosses of the FGF10^{fl/fl}/Flrt1/3^{fl/fl} mice also with the Emx1 Cre or vice versa – Cep83 also with Foxg1-Cre.

6) Figure 6: how many replicates were used? Without replicates small differences such as the 60:40 versus 45:55 proportion in cells destined to express Flrt3 may not be significant and relevant.

7) The authors focus on embryonic time points in their analysis, but it would be important to know if the folds are transient or persist.

Version 1:

Reviewer comments:

Reviewer #1

(Remarks to the Author)

The authors provided additional experimental data and explanation in their revised manuscript. Although I recognize the effort they put into making revisions, I am afraid that there are still critical points the authors should address.

Major points

1) In response to my previous major point #1, the authors wrote the following in the rebuttal letter:

"The novelty of our study is that we have combined these two mechanisms in a single animal and found that the combination produces a synergistic effect that goes beyond their individual effects. Moreover, we show that the formation of gyri and sulci involve different mechanisms, with specific cortical progenitors favouring each one."

However, in the rebuttal letter, the authors did not explain which data showed synergistic effects rather than additive effects. If synergistic effects exist, the values should be statistically significantly larger than those that would be obtained with additive effects. The authors should provide the data showing the synergistic effect they mentioned.

2) In my previous major point #2, I wrote the following:

"In Figures 2 and 4, the authors counted progenitor cells using dKO and tKO mice. The results seemed similar to those previously obtained using Cep83-KO and Fgf10-KO mice, and it was unclear what new information these results add compared to the previous results. It would be intriguing to examine the synergistic effects of Cep83-KO and Flrt1/3-KO on progenitor numbers, and those of Fgf10-KO and Flrt1/3-KO on progenitor numbers."

However, the authors did not deal with this point in their rebuttal letter. The authors should address my previous comment.

3) As I wrote in my previous major point #3, I was still not convinced of the authors' idea that these protrusions are physiological folds rather than cortical heterotopia because the authors failed to show curvature both at the cortical surface and at the border between gray and white matter. In gyrencephalic animals, curvature in both places appears at the same time during development. The authors should provide additional data clearly showing that their animals exhibit the features of actual cortical folds.

4) The authors proposed that expansion of IP cells leads to sulcus formation, whereas expansion of apical progenitors results in gyrus formation. Therefore, in my previous major point #4, I suggested that the authors show that IP cells and apical progenitors are abundant in sulcal regions and gyral regions, respectively, in developing gyrencephalic brains. In response to my comment, the authors provided Supplementary Fig. 6d, but it only showed the relative abundance of IP cells and bRG cells. Previous studies showed that IP cells were more abundant in the future gyral regions than in the future sulcal regions in the developing gyrencephalic brain. Thus, it is inappropriate to conclude that expansion of IP cells leads to sulcus formation. The authors' data indicates that the relative ratio of IP cells and bRG cells determines where gyri and sulci will be formed. The authors should change their descriptions throughout the manuscript to clearly reflect this.

5) In response to my previous minor point #3, the authors added separate staining images in Supplementary Fig. 1i-k and said cortical layers were preserved. The separate images were helpful, but the cortical layers did not seem to be well preserved. Is there layer 4 in Supplementary Fig. 1j, k?

6) In response to my previous minor point #5, the authors examined the number of bRG cells by counting the number of Pax6/PH3-double positive cells (new Figure 2f-j, Supplementary Fig. 2f, Supplementary Fig. 4e,f). However, Pax6/PH3-positive cells represented only dividing bRG cells rather than all bRG cells. The authors should count the number of Pax6-positive/Tbr2-negative cells in the SVZ to examine the number of bRG cells.

Minor points

1) Although the authors wrote as if Satb2 is an upper layer marker, Satb2 is expressed in callosally projecting neurons, which are also abundantly distributed in deep layers. The authors should modify their description of Satb2 throughout the manuscript.

2) In response to my previous minor point #6, the authors added separate images (Supplementary Figure 5a). In the figure

legend, although the authors wrote "Boundaries of upper and deeper layers are indicated", there were 3 kinds of lines, and it was unclear which was the boundary. The authors should describe what each of the line types represents.

3) As the antonym of "upper", the authors used both "deeper" and "lower" in the text. The authors should use "lower" and remove "deeper".

Reviewer #2

(Remarks to the Author)

The authors have done a very nice job at addressing most of my comments on the previous version of this manuscript, and to my eyes their changes have improved the manuscript. However, and enthusiastic as I am about this study, I must insist in a couple of important issues in the manuscript that have not been properly addressed by the authors. I am sure that the authors will have no problem in addressing them, and thus my questioning must not be taken as a negative to have the manuscript published in Nat Comm; rather the opposite, I believe that the data of this important study must be presented with the best possible quality to later not be questioned once published.

1- The authors have analyzed Pax6+PH3+ cells as suggested, but they counted also cells inside the VZ dividing non-apically. These cells are not bRG but subapical progenitors, as described in Pilz et al. 2013 (Nat Comm). While subapical progenitors are rare in the mouse cortex, since the authors did not distinguish them from real basal progenitors outside of VZ, it remains possible that these are the cells that increase in the triple mutants, instead of bRGs.

2- Very importantly, the co-localization images are very far from convincing (actually some cells in several images of Gi 2 and Suppl Fig 2 are quite convincingly NOT double positive for Pax6/PH3, but indicated with arrowheads that they are) so I am not sure how were these could be analyzed and counted properly. The authors need to show some example detail images at higher magnification and proper z-resolution (i.e. single-plane confocal or apotome images).

3- Again on the identification of bRG, I had previously indicate that bRG are recognized also by pVIM stain, but where pVIM labels any cortical mitotic cell (as shown by multiple authors previously) while bRGs are identified by the presence of a basal pVIM+ process. To support increase in bRG, the authors have now analyzed Pax6+/PH3+ cells outside of VZ. However, as mentioned in my second point now, the images they present are very far from convincing. Showing analysis of pVim+ cells with and without a basal process would definitely strengthen the central conclusion of this study that bRG abundance increases in their mutants.

Version 2:

Reviewer comments:

Reviewer #1

(Remarks to the Author)

The authors improved their manuscript significantly. However, they should address the following points before publication.

1) In my major point #2 in the first decision letter, I wrote as follows: "In Figures 2 and 4, the authors counted progenitor cells using dKO and tKO mice. The results seemed similar to those previously obtained using Cep83-KO and Fgf10-KO mice, and it was unclear what new information these results add compared to the previous results. It would be intriguing to examine the synergistic effects of Cep83-KO and Flrt1/3-KO on progenitor numbers, and those of Fgf10-KO and Flrt1/3-KO on progenitor numbers". However, the authors did not perform additional experiments to examine these synergistic effects, and therefore, it remained unclear how the information in Figures 2 and 4 adds to our conceptual understanding. The authors should explain this point, which does not necessarily have to be related to synergistic effects.

2) As I wrote in my major point #3 in the previous decision letter, physiological folds are characterized by curvature at both the cortical surface and the border between gray and white matter, and curvature in both places appears at the same time during development. Although the authors wrote that previous papers used the term "folds" and "folding" even in the cases in which curvature appeared only at the cortical surface, it does not necessarily mean that the curvature in the previous papers really corresponded to genuine cortical folds. The authors should at least write that the phenotypes presented in this paper only partially recapitulate the features of physiological cortical folds, and it should be confirmed that the curvature truly corresponds to genuine cortical folds in future experiments.

Reviewer #2

(Remarks to the Author)

Unfortunately, and for some unexplained reason, the authors continue to name subapical progenitors as basal progenitors. This mistake is very clear now that they nicely show detailed images: in Fig 2g the arrowhead points at a basal mitosis (border between Pax6+ VZ and Pax6- SVZ), but in Fig 2h the arrowhead points at a subapical mitosis (middle of the Pax6+ VZ). Hence, this is an incorrect account of basal mitoses (and potential basal bRG), and does not solve my first remaining issue. The authors, in spite of now stressing that they quantify "Pax6/PH3-double positive cells", the section begins

indicating that “We estimated the number of dividing bRG cells characterized by expression of PH3 and Pax6 by cells outside the VZ”. The naming and quantifications of Pax6+/PH3+ cells must be done and presented properly, or else the result is misleading.

The authors nicely looked into pVim+ cells with a basal process, as I suggested previously, and identified such cells outside of VZ in some Cep83 tKO embryos, as shown in Fig 3 of the rebuttal letter. Given the above controversy about the identification of bRG with Pax6 and PH3 stains, these results should be included in the manuscript in that same section, to further support the increased presence of bRGs. Of note, the histogram shown in Fig 3 of the rebuttal letter seems somehow wrong, as in mutants there are individual datapoints at 0 and other at 1, so the average is not 1 as represented by the bars.

Version 3:

Reviewer comments:

Reviewer #1

(Remarks to the Author)

I do not have additional comments.

Reviewer #2

(Remarks to the Author)

Response to reviewers

Cortex Folding by Combined Progenitor Expansion and Adhesion-Controlled Neuronal Migration

Nature Communications manuscript NCOMMS-24-35775-T

We would like to thank the reviewers for their careful reviews and comments which have helped us to further improve the manuscript. We have responded with a substantial amount of additional data and analysis, added more clarification where needed, and we revised text and figures accordingly. Changes in the main text are indicated in red.

Reviewer #1:

Chun et al. investigated the mechanisms underlying cortical folding using various kinds of mice. They examined the phenotypes of Cep83/Flrt1/3-tKO and FGF10/Flrt1/3-tKO mice and found that these mice exhibited cortical fold-like structures. They proposed that expansion of IP cells leads to sulcus formation, whereas expansion of apical progenitors results in gyrus formation. Although their findings are potentially interesting, and their data sets seem to be based on a huge amount of experiments, their findings unfortunately did not seem sufficient for significantly deepening our understanding of the mechanisms underlying cortical folding.

The authors nicely showed that Cep83/Flrt1/3-tKO and Fgf10/Flrt1/3-tKO mice exhibit cortical fold-like structures. However, previous reports have already shown that Cep83-KO, Fgf10-KO and Flrt1/3-KO mice also have fold-like structures. Although the phenotypes of Cep83/Flrt1/3-tKO and Fgf10/Flrt1/3-tKO mice were much stronger than Cep83-KO, Fgf10-KO and Flrt1/3-KO mice, the authors' findings do not seem to lead to a significant conceptual advance of our understanding of the mechanisms underlying cortical folding.

We thank the reviewer for this comment and for the opportunity to clarify this point. Indeed, previous reports focusing on cell proliferation (*Cep83* cKO, *Fgf10* cKO) or cortical migration (*Flrt1/3* dKO) have demonstrated fold-like structures in the typically smooth mouse cerebral cortex. However, the relative contributions of progenitor expansion and neuronal migration to the complex process of cortex folding have remained unclear. The novelty of our study is that we have combined these two mechanisms in a single animal and found that the combination produces a synergistic effect that goes beyond their individual effects. Moreover, we show that the formation of gyri and sulci involve different mechanisms, with specific cortical progenitors favouring each one. We have changed the text of the manuscript to highlight the novelty of our story (line 120-123 and 368-369).

In Figures 2 and 4, the authors counted progenitor cells using dKO and tKO mice. The results seemed similar to those previously obtained using Cep83-KO and Fgf10-KO mice, and it was unclear what new information these results add compared to the previous results. It would be intriguing to examine the synergistic effects of Cep83-KO and Flrt1/3-KO on progenitor numbers, and those of Fgf10-KO and Flrt1/3-KO on progenitor numbers. In addition, because the authors'

data showed that cortical folding was more prominent in the rostral cerebrum than in the caudal cerebrum, it would be intriguing to compare the increase in progenitor cells between rostral and caudal forebrains and examine the correlation between the degree of cortical folding and the degree of increase in progenitor cells.

This is a very good point, and we have now analyzed the correlation between the degree of cortical folding, measured as the gyrification index (GI), and the increase in progenitor cells from rostral to caudal regions. Additionally, we have included new quantifications in more dorsal regions for the *Cep83* tKO and intermediate regions for the *Fgf10* tKO models. Our data shows that the regions with the highest GI in both the *Cep83/Flrt1/3* tKO and *Fgf10/Flrt1/3* tKO models present the strongest increase in progenitors (Supplementary Fig. 4I). In these regions, both tKO mouse lines exhibited an increased number of progenitors compared to their respective dKO lines and/or the previously reported values for single KO cells. For example, in lateral-rostral regions, *Cep83/Flrt1/3* tKO mice showed a 52% increase in the number of Tbr2⁺ intermediate progenitors compared to its *Cep83* dKO line, and more than a twofold increase compared to its CTL (*Flrt1*^{-/-}; *Flrt3*^{lox/+}) control (Fig. 2b,c), which is higher than the previously reported 30% increase in the single KO line ¹. Similarly, in *Fgf10/Flrt1/3* tKO mice, rostral regions showed a 30% increase compared to its control (Fig. 4b).

Physiological folds exhibit curvature not only at the cortical surface but also at the border between gray and white matter. The authors should examine if the border between gray and white matter also shows curvature in dKO and tKO mice. The pictures in Figures 3K, 3L and 5B showed the curvature at the surface but not deep in the cortex. Figures 3K, 3L and 5B seemed to show heterotopia rather than folds.

The images showing folds in Figures 3 and 5 are from E17.5 *Fgf10/Flrt1/3* tKO embryos. At this developmental stage, it is too early to examine the border between gray and white matter because cortical gliogenesis begins around E16-E17 ³. We agree with the reviewer that the folds in these figures, corresponding to the *Fgf10/Flrt1/3* tKO embryos, do not extend into the deeper cortical layers as seen in the *Cep83/Flrt1/3* tKO model (Fig. 1). This difference is consistent with each model favoring either gyri-like (*Fgf10* tKO) or sulci-like (*Cep83* tKO) structures. These observations are in line with other studies on gyrencephalic animals, where folding in deeper layers is observed in sulci ⁴, but not in gyral areas where the white matter tends to protrude ⁵.

The authors proposed that expansion of IP cells leads to sulcus formation, whereas expansion of apical progenitors results in gyrus formation. Is this consistent with the data from gyrencephalic animals? The authors should show that IP cells and apical progenitors are abundant in sulcal regions and gyral regions, respectively, in developing gyrencephalic brains.

Thank you for this suggestion, which we have followed up by examining the abundance of apical (Pax6) versus intermediate (Tbr2) progenitor expression markers in the prospective splenial gyrus

(SG) and lateral sulcus (LS) using recently published RNA-seq data from ferret ⁶. We used the same metadata and cluster categorization provided by the authors to quantify the expression of the apical (Pax6) and basal (Tbr2) gene markers in the germinal zone (average expression from VZ+SVZ) of both gyri and sulci. We then calculated the expression proportion between these two markers for both regions. Consistent with our results, we found an increased ratio of Pax6 to Tbr2 expression in the gyrus (SG) compared to the sulcus (LS) (Supplementary Fig. 6d).

In Figure 6F, the authors showed that Cep83-tKO mice had increased upper layer neurons, while the authors concluded that Cep83-tKO leads to sulcus formation. Are upper layer neurons increased at sulci in gyrencephalic animals?

The results shown in Fig. 6f indicate that the proportion of upper layer neurons is higher compared to the deeper layers in the *Cep83* tKO model. This finding is consistent with the reduced thickness of deeper cortical layers observed in the sulcus areas of this model (Fig. 5g-j) and in the sulci of the *Flrt1/3* dKO⁷, leading to reduced numbers of Ctip2+ deeper layer neurons (Supplementary Fig. 5a-c). These results align with the reduced thickness of deeper cortical layers observed in the sulci of gyrencephalic species such as the ferret ⁴.

In Figure 6L-R, the authors tried to provide a theoretical basis for distinguishing gyrus and sulcus formation. I understood that these parameters affect the morphology of folds, but it was difficult to distinguish gyri and sulci from their modeling. The authors should provide more convincing modeling data sets.

Thank you for suggesting to clarify and improve our modeling data. The 3D contours of Fig. 6n-q illustrate the distribution of particles along the Z-axis after computer simulations, color-coded from blue to red based on each particle's Z position. In the control situation (CTL), the surface appears uniform with little dispersion of particles along the Z-axis, as depicted in the 2D contour. In contrast, the different mutant conditions produce a wavy surface with increased dispersion along the Z-axis of the particles (Fig. 6o,p). Specifically, in the condition representing the *Fgf10/Flrt1/3* tKO (Fig. 6q), some particles reach very high Z positions (above 1.2), which represent elevated areas mimicking the protrusion/gyri-like structures that we see in this model. We have colored the areas representing gyrus-like structures in red in the 2D contour to better illustrate this in the figure. We also provide a new quantification of the percentage of simulations generating these structures in the different models. A new graph in Fig. 6r (bottom) shows that only in the condition representing the *Fgf10/Flrt1/3* tKO do we obtain gyrus-like structures in 76% of the cases, which contrasts with the 4-8% in other conditions.

Regarding the sulci, we considered the deeper areas of the surface (below 0.5) as sulcus-like structures and colored them in blue to better illustrate their appearance in conditions that mimic the *Flrt1/3* dKO and *Cep83* tKO situations (Fig. 6o,p). We have updated the figure legend and results text accordingly (line 354-356).

Minor comments:

The authors showed the penetrance and the GI in Figures 1 and 3. In addition to this, it would be intriguing to examine the number of folds per section per each animal.

The number of folds per section varies significantly between animals, as we previously showed for FLRT1/3 DKO ⁷. We observed similar variability in the tKO mutants, with on average around 1-2 folds per section extending from rostral to intermediate sections. Occasionally, we find multiple folds in one section, as shown in Fig. 1g (up to 8). To the best of our knowledge, such examples are not observed in any other model. This variability could be linked to the absence of gene expression microdomains in the mouse brain. Several studies have shown that these microdomains are present at multiple locations across the developing cortex of gyrencephalic species such as the ferret, but not in the lissencephalic mouse ⁸. Notably, the penetrance of the folds is very high and consistent in these mutants (around 90%), which contrasts with the lower penetrance observed in dKO or single mutants (Fig. 1l and Fig 3n).

Fig 1E seemed incomplete. Some bars (e.g. median, whisker) seem to be missing.

Thank you for pointing this out. We have updated the Y-axis of this panel to better showcase the median line that was previously missing in the *Cep83* dKO group. The absence of a visible median line in the *Cep83* dKO group was due to its value being close to the 25th (or 75th) percentile.

Although the authors wrote that cortical layers were preserved (page 5, third line from the bottom; page 7, second line from the bottom), it was difficult to distinguish cortical layers in the pictures. The authors should provide more convincing pictures showing all cortical layers and the border between gray matter and white matter. Separate immunostaining images rather than merged images would be better

Agreed, we have included separate stainings in Supplementary Fig. 1i-k to better illustrate the upper and lower cortical layers. We have now complemented this data with new images from folds found in postnatal stages for the *Cep83* and *Fgf10* tKO lines to demonstrate that these are not transient embryonic structures and that cortical layering is preserved (Supplementary Fig. 5j-m).

The names of genes should be consistent. Different formats were used: "TBR2/Tbr2," "pH3/PH3," "pVIM/Pvim," "Sox2/SOX2," "Satb2/SATB2," "FLRT/Flrt" etc.

Thank you and agreed, we have revised all gene names to Sox2, Tbr2, PH3, Pvim and Flrt.

In the text and in Figures 2E and 2F and Extended data Figure 4E, the authors counted bRG cells by examining the number of pH3+/pVim+ cells. However, pH3+/pVim+ cells represent a subset of bRG cells. The authors should use appropriate markers for counting bRG-like cells.

Thank you for pointing this out. As also suggested by reviewer #2, we have complemented our results by quantifying double-positive Pax6 and PH3 cells located more than 60µm (approximately three nuclei) above the apical surface (outside the VZ) (Fig. 2f-j, Supplementary Fig. 2f, Supplementary Fig. 4e,f). These results revealed an increased number of basal Pax6/PH3 cells in E13.5 *Cep83* tKO cortex compared to both the *Cep83* dKO and control, and a reduction in the case of E11.5 *Fgf10* tKO cortex compared to controls. We have added this data to the results section (line 174-176 and 234-235).

The authors drew dashed lines to indicate the borders between cortical layers, but the reasons why they were able to draw the lines at specific positions were often unclear. For example, in Figure 5H, the border between the uCP and the iCP and that between the iCP and white matter seemed unclear. The authors should provide clearer evidence.

Agreed. We now provide separate stainings for cortical plate layers: *Satb2* (upper) and *Ctip2* (deeper) in Supplementary Fig. 5a to clearly delineate the borders between upper and lower cortical plate.

*The authors should confirm that apoptosis did not occur in sulcal regions in *Cep83/Flrt1/3-tKO* and *FGF10/Flrt1/3-tKO* mice.*

Thank you for this suggestion. We have not detected increased cell death in the sulcus and gyrus regions areas in either of these two models by performing caspase 3 staining at E17.5 (Supplementary Fig. 5d-g). We added this data to the results section (line 262-263).

It seemed that "SATB+" should be "SATB2+" in Figure 5C and 5F.

Thank you for detecting this typo, we have updated the Y legend of both panels to "Number of *Satb2* cells per 150um column"

Reference #52 did not have enough information.

Thanks for noticing this, we have added a suitable reference.

Reviewer #2:

This is a fascinating study from the Klein lab, building on their previous findings about mechanisms of cerebral cortex folding. They previously showed that specific mechanisms and patterns of neuron migration are sufficient to drive folding of the mouse cortex, and here they show that

combining the effect of neuron migration with increased cell proliferation exacerbates the cortex folding phenotype. The authors combine their previous mutant mouse model (Flrt1/3 dKO) with two additional conditional mutant mouse lines where progenitor proliferation is enhanced (Cep83 KO, Fgf10 KO). In both triple mutants, they show very robust and abundant folding, much more than in the separate mutant conditions. Intriguingly, however, in one case (Cep83) they show the seeming formation of sulci, whereas in the other (Fgf10) they show the formation of gyri. This is fascinating because then they show that the immediate effect of these mutations is different in each case, thus suggesting the existence of distinct and complementary mechanisms for the formation of folds and fissures in gyrencephalic mammals. The study ends with an analysis of cell types with altered abundance in each mutant condition, and a mathematical model aimed at supporting a hypothesis on how the multiple changes found in these mutants may collectively explain and contribute to the development of cortical folds. In my opinion, this is a perfect fit for Nature Communications and I am convinced that it will become a reference paper in the field. I would only recommend to tackle a handful of formal aspects, as detailed next.

We are grateful for this reviewer's accurate and supportive review. We respond to individual points in full details below.

The authors conclude their Intro by stating "Our results identify key developmental mechanisms that collaboratively drive cortex folding". This should be adjusted, as proliferation and cell migration had already been experimentally demonstrated as mechanisms that drive cortex folding, and their interaction/synergy in this process had been proposed. The novelty of this study is to demonstrate that the combination of these mechanisms enhances their effect on folding beyond the individual effects, and that the formation of gyri may involve slightly but significantly different mechanisms from the formation of sulci.

We are thankful for the reviewer's comments and have adopted the wording of the introduction, which now reads: "Our results reveal that the combination of increased proliferation and divergent cell migration enhances their effect on cortex folding beyond the individual effects. Moreover, we show that the formation of gyri involves slightly, but significantly different mechanisms from the formation of sulci". (line 120-123)

In the first paragraph of page 6, the authors indicate that Cep83 tKO leads to a strong cortex folding phenotype with the presence of bilateral sulci. However, it is not shown nor demonstrated that any sulci are bilateral in these mutants. The observations should be referred to as "folding phenotype in both cerebral hemispheres".

Agreed, we have rephrased this sentence to indicate that sulci occur in both cerebral hemispheres.

Last paragraph of page 6: "we estimated the number of PH3+ basal cells stained for phosphovimentin (pVIM), which is known to be expressed in RG cells during cell division". While this is true, phVim stain is not specific to aRG but to all mitotic progenitor cells in the cerebral cortex, namely including IPCs. The standard criteria to identify bG are positivity for PH3 (or phVIM) and Pax6 outside of VZ. Thus, these observations must be rephrased and the conclusions re-considered.

We thank the reviewer for highlighting this point. We have quantified double-positive Pax6 and PH3 cells located more than 60µm (approximately three nuclei) above the apical surface (outside the VZ) (Fig. 2f-j, Supplementary Fig. 2f, Supplementary Fig. 4e,f). These results revealed an increased number of basal Pax6/PH3 cells in E13.5 *Cep83* tKO cortex compared to both the *Cep83* dKO and control, and a reduction in the case of E11.5 *Fgf10* tKO cortex compared to controls. We have added this data to the results section (lines 174-176 and 234-235).

“Collectively, these results indicate that deletion of Cep83 causes an expansion of IPs and bRG cells at E13.5” - As mentioned in the previous point, to support this conclusion the authors must show increases in basal PH3 that are positive for Tbr2 (IPC) or Pax6 (bRG). Unfortunately, their current results only show increased Tbr2+ cells (not Pax6!), and then increased basal proliferation but with no cell type-specific markers. pVIM labels any cortical mitotic cell, as shown by multiple authors previously, while bRGs are identified by the presence of a basal pVIM+ process.

Thank you for this important comment. As discussed in the previous point, we have performed new experiments to quantify bRG cells as double-positive Pax6/PH3 cells outside the VZ. As shown in Fig. 2f-j, Supplementary Fig. 2f, Supplementary Fig. 4e,f, we found an increased number of these cells in *Cep83* tKO embryos.

In Fig 2, the authors indicate that quantifications were done on the most lateral cortex (almost ventral), whereas the folding phenotype of these mutants is most obvious in parietal and dorsal regions. These quantifications should be re-done in the proper areas to proof the point that such changes in proliferation are related to such gyrus formation. Second, given the concepts of patterned proliferation and gene expression correlated with folding, mentioned in the Intro, does proliferation play a specific pattern along the latero-medial axis of the embryonic cortex of these mutants, which would explain the formation of these sulci?

We thank the reviewer for their insightful comments. We had analyzed the most lateral part of the cortex in rostral sections because we found several sulci in those locations at E17.5 (Fig. 1h,j and Fig. 5e). We agree with the reviewer that some embryos exhibited sulci in more dorsal locations (Fig. 1i). Therefore, we have complemented the results in Figure 2 with quantifications from these regions. The new results show an increased number of intermediate progenitors (Tbr2+) in these regions in *Cep83* tKO embryos (Fig. 2d,e, Supplementary Fig. 2d,e).

Regarding the link between proliferation and folding, we have analyzed the correlation between the degree of cortical folding, measured as the gyrification index (GI), and the increase in progenitor cells. Our data shows that the regions with the highest GI in both the *Cep83* tKO and *Fgf10* tKO models present the strongest increase in progenitors (Supplementary Fig. 4i).

Why Fgf10 Ix/Ix mice had 10% folding? Aren't these control embryos?

Thank you for this comment. Indeed, *Fgf10* Ix/Ix embryos serve as littermate controls for the *Fgf10* cKO. To maximize the chances of obtaining *Fgf10* cKO embryos, the mother was *Fgf10* Ix/Ix and carried the Foxg1-Cre. One possible explanation for observing folding in two embryos could be germline effects of this Cre line. If recombined in the germline, the floxed allele from the mother

may in fact be a null allele, and the *Fgf10* lx/lx offspring may in fact be heterozygous mutant. The degree of folding was very mild in these embryos, characterized by a slightly wavy cortical plate. We have added this explanation in the figure legend of Fig. 3n.

Panoramic views in Fig 3J fail to reveal folds or fissures; should be replaced by examples where this is somewhat visible.

Agreed, we have replaced the images with new panoramic views that clearly show the presence of macroscopic gyri (Fig 3j).

Thickness of Satb2 layer is clearly larger in mutants than controls. Does this also affect the Ctip2 layer? This may be expected given the early recombination by FoxG1-Cre.

Indeed, our data show that the number of upper cortical neurons (Satb2+) is increased in the gyrus areas of *Fgf10*tKO (Fig. 5h). We have quantified the number of Ctip2+ cells at this location relative to the adjacent and contralateral sites and did not find differences (Supplementary Fig. 5 c), which aligns with the consistent thickness of this layer as shown in Fig. 5h. This result contrasts with the *Cep83*tKO embryos, where in sulcus areas the Ctip2 layer is much thinner (Fig. 5g) and contains fewer Ctip2+ neurons (Supplementary Fig. 5b).

The authors report an expansion of Sox2+ apical progenitors in Fgf10 dKO and tKO mice compared to littermate controls (Flrt1-/-;Flrt3lx/+). The views displayed in Fig. 4A clearly show that this analysis was done in the most rostral part of the telencephalon (no basal ganglia nor thalamus visible). Analysis of this region in the coronal plane is bound to high variability, because sections are far from perpendicular to the tissue and thus cortical thickness (and cell number) varies much from one section level to the next. As shown in Figure 1, the cortex folding phenotype extends from rostral to caudal cortex in these mutants, so the authors can choose where is most reliable to do the analysis. For these reasons, quantifications should be done in a more intermediate level of the telencephalon.

We thank the reviewer for these comments allowing us to clarify our findings. Figure 1 shows the cortex folding phenotype of *Cep83* tKO embryos, which indeed extends from rostral to caudal regions. Figure 3 shows the phenotype of *Fgf10* tKO embryos which is, however, restricted to rostral regions, because *Fgf10* expression is confined to the rostral cortex and *Fgf10* deletion had previously been shown to cause expansion and thickening only of the rostral cortex². We have changed the text to make this clearer for the reader (lines 195-197).

The original Figure 4 presented quantifications in the most rostral cortical regions of *Fgf10* control and mutant embryos. We have now added quantifications at a more intermediate level of the cortex confirming that the expansion of Sox2+ apical progenitors is primarily seen in rostral regions (Fig. 4c,e, Supplementary Fig. 4d).

The results presented in Figure 5 are quite confusing. First, Ctrl embryos have very few apical mitoses, which concomitantly leads to a 50-50% proportion apical-to-basal. This is very much unlike previous observations from many different labs at this same developmental age, and

should be revised. Second, dKO and tKO have a clear increase in Tbr2+ cells as shown in the example images, but this is not reflected in the quantification of Fig 4B; in addition, dKOs and tKOs virtually lack basal mitoses, which is particularly dramatic in tKOs. As mentioned in a previous comment, Tbr2+ cells really newborn neurons? IPCs stalled somewhere in the cell cycle? If ablation of Fgf10 expands apical progenitors, why are apical mitoses not significantly increased in tKOs?

We thank the reviewer for their careful review of the data. The reviewer is correct. Under control conditions, the number of apical mitoses should be much higher in controls. We have added more samples, re-quantified the data, and found that the proportion apical-to-basal in controls is 80 to 20% consistent with previous studies (Fig. 4d). We have replaced the image for PH3+ cells in controls to better match the quantified data (Fig. 4a).

Consistent with previous studies⁹ we found that the majority of Tbr2+ cells (80%) are proliferative progenitors, as demonstrated by a short BrdU pulse (Supplementary Fig. 4g,h).

In page 10: “Cep83 tKO embryos also showed an increased proportion of upper layer neurons” - The difference is very mild, and to draw such conclusion one should test statistical difference

We understand that the results referred to are those shown in panel Fig. 6f, where we observed an increased proportion of upper cortical neurons in the *Cep83* tKO by analyzing the populations identified by scRNA-seq at E15.5. Even though the increase is around 10%, it is statistically significant, as analyzed by the chi-square test (Fig. 6f). This result is consistent with the increased proportion of upper cortical neurons present in this mouse model at later stages, as shown by long BrdU pulses (Fig. 5k-l).

Page 13: “This would be consistent with studies showing that in gyrencephalic species, regions in the cortical plate with lower and higher neuronal densities correspond to future sulci and gyri, respectively, during development”. - Directly related to these observations are some questions very relevant to understand the underlying mechanisms: are there differences in proliferation, IPC or bRG abundance along the developing cortex of these mutant mice? If yes, do these spatial differences explain the patterns of gyri/sulci? Are the trajectories of radial glia fibers modified as to explain the formation and location of gyri/sulci? The authors should address these questions, or at least discuss them with due consideration.

These are excellent questions, which we have addressed with new data. Regarding the differences in proliferation and the patterns of gyri/sulci, we have now analyzed the correlation between the degree of cortical folding, measured as the gyrification index (GI), and the increase in progenitor cells. Our data shows that the regions with the highest GI in both the *Cep83/Flrt1/3* tKO and *Fgf10/Flrt1/3* tKO models present the strongest increase in progenitors (Fig. 1a,b). In these regions, both lines exhibited an increased number of progenitors compared to their respective dKO lines and the previously reported values for single KO cells (Supplementary Fig. 4i).

Moreover, we have performed BLBP staining to visualize the trajectories of radial glia fibers. We found that their trajectories converge in the sulcus areas of the *Cep83* tKO and diverge in the

gyrus areas of the *Fgf10* tKO which is similar to those reported in classic descriptions of gyrencephalic species such as ferrets and monkeys^{4,10} (Supplementary Fig. 5h,i).

Reviewer #3:

*In this manuscript the authors cross several mouse lines known to elicit a phenotype in cortex folding to examine possible additive or synergistic effects. The rationale is that deletion of *Flrts* affects mostly migration and adhesion/repulsion, while the *Cep83* or *FGF10* deletion affects stem progenitor cells, such that affection both may cause more severe effects. Indeed, this is the case in triple mutants for *Cep83* and *Flrt1/3* with impressive folding (mostly sulci formation) in the triple mutants. The authors further demonstrate an accompanying increase in basal mitoses and *Tbr2*⁺ cells, i.e. basal progenitors, but not apical stem/progenitor cells. If this is causative for the more severe folding phenotype has not been determined. The second set of data crossing the *Flrt1/3*dKOs with *FGF10*cKO is less convincing, in particular in regards to what is referred to as “a fold” (see below). This is the weak point, as it casts doubts on the interesting conclusion that *Cep83* deletion in combination with *Flrt1/3* affects selectively sulci versus the combination with *FGF10* deletion affecting gyri formation. The phenotypes are clearly different, but if the condensates of cells observed in the *FGF10* deletion together with *Flrt1/3* are really gyri is highly doubtful. Moreover, the authors focus on a single time point, collecting embryos just prior to birth, but it would be very important to know about later phenotypes – are they persisting or transient in nature. Taken together, this is a highly promising manuscript that could become a broadly important piece of work if the questions below can be answered.*

We thank the reviewer for their insightful and constructive comments. We are in full agreement and have answered all criticisms in detail.

For all histograms please show the data points in the histogram and provide information about the number of embryos analyzed from how many different litters (e.g. Figure 2, 4, 5, Figure S5).

Agreed, we have revised these figures to provide all data points and the number of embryos/litters analyzed.

*In Figure 3C the authors outline a fold with a dashed line. However, there are clearly *Satb2*⁺ cells on top of the line in the lower left corner, raising doubts about this classification.*

Thank you for detecting this mistake. We have redrawn the dashed line to more accurately outline the shape of the fold.

In Figure 3J the macroscopic sulci and gyri are not visible. Please provide better examples, if they exist.

Agreed, we have replaced the images with new panoramic views that show the presence of macroscopic gyri (Fig. 3j).

Also in Figure 4K there is largely a dark space in SATB2-staining between the alleged “fold”, but these may be weaker labelled SATB2+ cells (as suspected from the example in 3C), or if not, which cells are there? Other neurons/no neurons? There is a huge black space between CTIP2+ band and the pial surface that is very smooth – what is in this black hole?

Thank you for suggesting to clarify this. The black space present in Fig. 3k above the Satb2+ cells represents a thicker marginal zone because the basal membrane remains largely smooth. This aligns with previous observations in the *Flrt1/3* dKO embryos, where the marginal zone thickens when the cortical plate folds but not the basal membrane⁷. We have updated the images for this figure with higher magnification to show that there are DAPI+ cells in this area (Supplementary Fig. 3i), As requested by this reviewer further below, we have also added images of postnatal P1 brains. In one of these images of a *Fgf10* tKO brain, we have indicated a region between the gyri with absent Satb2 immunoreactivity. In this region there are DAPI+ cells (Supplementary Fig. 5I, image on the right with arrowheads). We have revised the figure legends to clarify that the marginal zone thickens.

It is understandable, but also problematic that the authors use different Cre driver lines for the Cep83 versus Fgf10 deletions. Understandable, because they use the lines that have been used before in the single cKOs, but problematic, because we formally do not know if the differences in phenotype may relate to a different onset in the timing of recombination. I therefore suggest to do some pilot crosses of the Fgf10fl/fl/Flrt1/3flfl mice also with the Emx1 Cre or vice versa – Cep83 also with Foxg1-Cre.

We thank the reviewer for this constructive criticism regarding the potential impact of the different Cre lines used to generate *Cep83* vs. *Fgf10* tKOs. Given that *Fgf10* expression is transient, peaking around E9-E11², we could not use the *Emx1-Cre*, which recombines at later time points (Supplementary Fig. 1a). Instead, we used the *Foxg1-Cre*, which offers earlier and more efficient recombination in the dorsal telencephalon where *Fgf10* is primarily expressed. We agree that using the *Foxg1-Cre* line for the *Cep83* tKOs would be a useful experiment. However, it would require several rounds of crossings and a careful analysis with comparable cohorts of animals and is beyond the scope of this study. We have noted this in the discussion as an open question for future research (lines 431-432).

Figure 6: how many replicates were used? Without replicates small differences such as the 60:40 versus 45:55 proportion in cells destined to express Flrt3 may not be significant and relevant.

Thanks for pointing this out. We used three independent samples for each tKO line and two controls for each mutant. We have added this information to the figure legend and confirmed a significant difference in the proportion of cells by the chi-square test (Fig 6i).

The authors focus on embryonic time points in their analysis, but it would be important to know if the folds are transient or persist.

We thank the reviewer for this important comment. We have added images and quantifications to the revised Supplementary Fig. 5j-m showing folds in postnatal P1 stages for the *Cep83* and *Fgf10* tKO lines. The observed anatomy of the folds is preserved (sulci in case of *Cep83* tKO and gyri in case of *Fgf10* tKO) and the penetrance is high (60% for *Cep83* tKO and 80% for *Fgf10* tKO). These results indicate that the folds persists into early postnatal life. Whether they persist into adulthood and have functional consequences will be an exciting project for the future. This was added to results and discussion (lines 373-374 and 432-434).

References

1. Shao W, *et al.* Centrosome anchoring regulates progenitor properties and cortical formation. *Nature* **580**, 106-112 (2020).
2. Sahara S, O'Leary DDM. Fgf10 regulates transition period of cortical stem cell differentiation to radial glia controlling generation of neurons and basal progenitors. *Neuron* **63**, 48-62 (2009).
3. Zhang X, *et al.* Clonal Analysis of Gliogenesis in the Cerebral Cortex Reveals Stochastic Expansion of Glia and Cell Autonomous Responses to Egrf Dosage. *Cells* **9**, (2020).
4. Smart IH, McSherry GM. Gyrus formation in the cerebral cortex of the ferret. II. Description of the internal histological changes. *J Anat* **147**, 27-43 (1986).
5. Rash BG, Tomasi S, Lim HD, Suh CY, Vaccarino FM. Cortical gyrification induced by fibroblast growth factor 2 in the mouse brain. *J Neurosci* **33**, 10802-10814 (2013).
6. Del-Valle-Anton L, *et al.* Multiple parallel cell lineages in the developing mammalian cerebral cortex. *Sci Adv* **10**, eadn9998 (2024).
7. del Toro D, *et al.* Regulation of Cerebral Cortex Folding by Controlling Neuronal Migration via FLRT Adhesion Molecules. *Cell* **169**, 621-635.e616 (2017).
8. de Juan Romero C, Bruder C, Tomasello U, Sanz-Anquela JM, Borrell V. Discrete domains of gene expression in germinal layers distinguish the development of gyrencephaly. *The EMBO Journal* **34**, 1859-1874 (2015).
9. Englund C, *et al.* Pax6, Tbr2, and Tbr1 are expressed sequentially by radial glia, intermediate progenitor cells, and postmitotic neurons in developing neocortex. *J Neurosci* **25**, 247-251 (2005).
10. Rakic P. Mode of cell migration to the superficial layers of fetal monkey neocortex. *J Comp Neurol* **145**, 61-83 (1972).

REVIEWER COMMENTS

Cortex Folding by Combined Progenitor Expansion and Adhesion-Controlled Neuronal Migration

Nature Communications manuscript NCOMMS-24-35775A.

We would like to thank the reviewers for their careful reviews and comments which have helped us to further improve the manuscript. We have responded with a substantial amount of additional data and analysis, added more clarification where needed, and we revised text and figures accordingly. Changes in the main text are indicated in red.

Reviewer #1:

The authors provided additional experimental data and explanation in their revised manuscript. Although I recognize the effort they put into making revisions, I am afraid that there are still critical points the authors should address.

1) In response to my previous major point #1, the authors wrote the following in the rebuttal letter: "The novelty of our study is that we have combined these two mechanisms in a single animal and found that the combination produces a synergistic effect that goes beyond their individual effects. Moreover, we show that the formation of gyri and sulci involve different mechanisms, with specific cortical progenitors favouring each one." However, in the rebuttal letter, the authors did not explain which data showed synergistic effects rather than additive effects. If synergistic effects exist, the values should be statistically significantly larger than those that would be obtained with additive effects. The authors should provide the data showing the synergistic effect they mentioned.

We agree with this reviewer that a statistical analysis of synergistic versus additive effects would strengthen the observations. We calculated the number of folds in each brain of the relevant cohorts and asked if the value of the triple knockouts was significantly larger than that expected from a mere additive effect. We analyzed all brain sections that were previously used to calculate the penetrance of folding. For the *Cep83* mutant mice, the average number of folds in controls (*Cep83* lx/lx; no Cre) was 0.1, in *Cep83* single KO was 0.5, and in *Flrt1/3* double KO was 0.3. We subtracted the control value (0.1) from all samples and calculated the additive prediction, which summed up to 0.6. This value is now shown as "Additive effect" in the new panel Fig. 1m. This additive prediction was then statistically compared to the number of folds observed in the combined *Cep83* dKO and *Cep83* tKO cohorts (mean number of folds 1.56 \pm SD and 1.83 \pm SD, respectively). We used the one-sample Wilcoxon test to assess the differences between the summed additive effect and the observed phenotype in the double and triple knockouts. The results revealed that the phenotypes in the combined lines were significantly greater than the theoretical additive effect (in both cases $p < 0.01$). This significant difference supports the presence of synergistic effects rather than mere additive interactions.

A similar calculation was performed for the *Fgf10* mutant lines. The mean numbers of gyri in the combined lines, *Fgf10* dKO and *Fgf10* tKO were 1.45 +/- SD and 1.77 +/- SD, respectively. These values were significantly greater than the theoretical additive effect (0.1; in both cases $p < 0.001$). The results are included in the new panel Fig. 3o. They strengthen our conclusion that the combination of increased proliferation and divergent cell migration enhances their effect on cortex folding beyond the individual effects.

2) In my previous major point #2, I wrote the following: "In Figures 2 and 4, the authors counted progenitor cells using dKO and tKO mice. The results seemed similar to those previously obtained using *Cep83*-KO and *Fgf10*-KO mice, and it was unclear what new information these results add compared to the previous results. It would be intriguing to examine the synergistic effects of *Cep83*-KO and *Flrt1/3*-KO on progenitor numbers, and those of *Fgf10*-KO and *Flrt1/3*-KO on progenitor numbers." However, the authors did not deal with this point in their rebuttal letter. The authors should address my previous comment.

We realized that we did not address this reviewer's point regarding the quantification of progenitor cells in single KO conditions. We agree that exploring these conditions could be interesting. The increase in *Tbr2*⁺ progenitors in the dorsolateral cortex was reported to be 25% in E15.5 *Cep83* single KO¹. We found increases of 64% in the lateral-medial cortex of E13.5 *Cep83;Flrt1* double KO and 114% in *Cep83;Flrt1/3* triple KO (Fig. 2c). We also analyzed E15.5 brains and found increases of 10% in the lateral cortex of *Cep83;Flrt1* double KO and 44% in *Cep83;Flrt1/3* triple KO (see Figure 1 below). It may be that there is a significant difference between E15.5 *Cep83* single and E13.5 combined *Cep83;Flrt* mutants. It may also be that the difference is largely due to the difference in embryonic stages. We have not re-analyzed the numbers of *Tbr2*⁺ progenitors in *Cep83* single KO embryos, because this genotype is not generated when breeding *Cep83* dKO/tKO mice and always had to be generated separately. Unfortunately, our single KO colonies are currently maintained at minimal levels. Expanding these colonies to perform the suggested experiments would require several months and the filing of an extension of our animal license. It is currently unclear, if the German authorities would grant this extension for an analysis that has previously been published.

Figure. 1: Quantifications of cell densities in the lateral area of rostral regions at E15.5 cortices of CTL, *Cep83* dKO and *Cep83* tKO. (CTL, *Flrt1*^{-/-};*Flrt3*^{lox/+} n = 6 brains, *Cep83* dKO n = 6 brains, *Cep83* tKO, n = 6

brains from 3 litters). Data are shown as mean \pm SEM; CTL vs. *Cep83* tKO $p = 0.0024$, $**p < 0.01$, one-way ANOVA with Tukey's post hoc analysis.

We suggest to leave this point unresolved and to mention the possibility of synergism on progenitor numbers in the manuscript. We state in the manuscript: "Collectively, these results suggest that deletion of *Cep83* causes an expansion of IPs and Pax6/PH3-double positive cells at E13.5 that results in an enlarged cortex at E17.5. Since this effect is already present in *Cep83* cKO mice and is only mildly enhanced, if at all, by the deletion of *Flrt1* and *Flrt3*, it follows that the expansion of IPs and Pax6/PH3-double positive cells in *Cep83* tKO embryos exacerbates the cortex folding phenotype seen in the *Flrt1/3* dKO mice." (lines 192-195)

In the case of *Fgf10*, a 30% increase in Pax6+ progenitors was reported in E14.5 rostral cortex of *Fgf10* single KO². We found a 25% increase of Sox2+ progenitors in E11.5 rostral cortex of *Fgf10* dKO/tKO embryos (Figure 4B). These results do not argue for a synergistic effect on progenitor numbers by combining all three alleles. For the same reasons as above, we did not re-analyze the numbers of Sox2+ progenitors in *Fgf10* single KO embryos. We have kept the conclusions regarding this point unchanged.

3) As I wrote in my previous major point #3, I was still not convinced of the authors' idea that these protrusions are physiological folds rather than cortical heterotopia because the authors failed to show curvature both at the cortical surface and at the border between gray and white matter. In gyrencephalic animals, curvature in both places appears at the same time during development. The authors should provide additional data clearly showing that their animals exhibit the features of actual cortical folds.

The kind of gyrus-like protrusions that we observed in *Fgf10* tKO animals have, to our knowledge, not previously been observed in mutant mice. In previous cases, protrusions were induced by electroporation, and curvature at the border between gray and white matter were not described³.

The kind of sulcus-like folds that we observed in *Cep83* tKO animals were previously described and curvature at the border between gray and white matter did not appear at the same time. For example, elevated Shh signaling by conditional expression of constitutively active Smoothed (SmoM2) resulted in cortical folding in P3 brains without curvature at the border between gray and white matter (see Supplementary Figure 4 of Wang et al., 2016, PMID: 27214567).

Supplementary Figure 4c of Wang et al., 2016

c. Nissl staining of coronal sections of control (*SmoM2loxP/+*) and *Nestin::CreER; SmoM2loxP/+* brains at P3. A relatively low dose of tamoxifen (1.5 mg/40 g of body weight, IP injection at E12.5) was used to avoid embryonic lethality. The arrows point to folds in the lateral cortices that are enlarged in the images on the right. Scale bar = 0.5 mm. Cortical folding was observed in approximately 30% of the *Nestin::CreER; SmoM2loxP/+* brains examined.

As we stated in our previous response, it is too early in mouse embryos to examine the border between gray and white matter because cortical gliogenesis begins around E16-E17⁴. We have added new images of a DAPI stained *Fgf10* dKO brain at postnatal stage P1 (new Supplementary Figure 5m) that show curvature at the cortical surface, while the white matter present in deeper layers remains smooth.

Another argument against cortical heterotopia is the fact that *Cep83* tKO and *Fgf10* tKO brains show normal layering and no mixing of cortical neurons during development (Supplementary Figure 1i-k and Supplementary Figure 3i, j) and postnatal stages (Supplementary Figure 5 j, k).

4) *The authors proposed that expansion of IP cells leads to sulcus formation, whereas expansion of apical progenitors results in gyrus formation. Therefore, in my previous major point #4, I suggested that the authors show that IP cells and apical progenitors are abundant in sulcal regions and gyral regions, respectively, in developing gyrencephalic brains. In response to my comment, the authors provided Supplementary Fig. 6d, but it only showed the relative abundance of IP cells and bRG cells. Previous studies showed that IP cells were more abundant in the future gyral regions than in the future sulcal regions in the developing gyrencephalic brain. Thus, it is inappropriate to conclude that expansion of IP cells leads to sulcus formation. The authors' data indicates that the relative ratio of IP cells and bRG cells determines where gyri and sulci will be formed. The authors should change their descriptions throughout the manuscript to clearly reflect this.*

We thank the reviewer for this comment. We may have unintentionally caused a misunderstanding. In Supplementary Figure 6d, we quantified the expression levels of the apical progenitor marker, Pax6, and the IP cell marker, Tbr2, within the germinal layers (SVZ+VZ) of

prospective gyrus and sulcus regions in the developing ferret cortex⁵. The ratio of Pax6 to Tbr2 expression in these regions (as shown in Supplementary Fig. S6d) suggests a higher relative abundance of apical progenitors in prospective gyrus areas than in sulcal regions.

In the same paper, the authors found a significant enrichment of Pax6 transcripts in the ventricular zone (VZ) of prospective gyrus (SG) compared to sulcus regions (LS) (Supplementary Figure S12 from Del-Valle-Anton et al., 2024 below). Similar findings were observed when we analyzed their published single-cell RNAseq data (Figure 2 below). This data was then used to generate the graph in Supplementary Figure 6d. This observation aligns with our findings in the *Fgf10* tKO mouse embryos, where at E11.5, regions destined to develop into gyrus-like structures by E17.5 exhibited an increased proportion of Sox2-positive apical progenitors compared to intermediate progenitors, as shown in Figures 4a-c of our manuscript. This data suggests that apical progenitors also participate in gyrus formation.

Supplementary Figure S12 E from Del-Valle-Anton et al., 2024: Expression patterns of *ZEB1* and *PAX6* in VZ and OSVZ in SG compared to LS.

Figure 2: Expression Patterns of Pax6 and EOMES in Prospective Gyrus and Sulcus Regions. Heatmaps showing the expression levels of Pax6 (left) and EOMES (Tbr2) (right) across different layers of the developing cortex in ferret embryos. Expression data is presented for the outer subventricular zone (oSVZ) and ventricular zone (VZ) within prospective gyrus (SG) and sulcus (LS) regions. Color intensity represents normalized expression levels, with red indicating higher expression and blue indicating lower expression. Pax6 exhibits higher expression in the VZ of prospective gyrus regions compared to sulcus. EOMES shows relatively uniform distribution. Data from Del-Valle-Anton et al., 2024 re-analyzed by us.

In case of *Cep83* tKO embryos, we found that combining the increased numbers of IP cells from *Cep83* cKO enhanced the sulcus-like folding phenotype observed in the *Flrt1/3* dKO when both lines were combined. Our cell transcriptomics and simulation data indicate that expansion of IP cells leads to an increased proportion of *Flrt3*-mutant neurons, which show increased lateral dispersion and favor sulcus-formation⁶. Therefore, we do not claim that an expansion of IP cells *per se* leads directly to sulcus formation. Instead, our data suggests that is the increased output of FLRT-mutant cells coming from these progenitors that favors the sulcus phenotype already present in the *Flrt1/3* dKO model.

We have modified sentences in the discussion to reflect these findings. "Expansion of intermediate progenitors by deletion of *Cep83* leads to an increase in *Flrt*-mutant neurons, thereby increasing cortex folding and the formation of sulci. Conversely, enlargement of the apical progenitor pool by deletion of *Fgf10* results in fewer *Flrt*-mutant neurons, which enhances cortex folding by promoting gyrus formation." (lines 381-384)

5) In response to my previous minor point #3, the authors added separate staining images in Supplementary Fig. 1i-k and said cortical layers were preserved. The separate images were helpful, but the cortical layers did not seem to be well preserved. Is there layer 4 in Supplementary Fig. 1j, k?

Thank you for this point. We now show low magnification images to illustrate that the layering in the folded regions is as well visible as the adjacent control regions (Supplementary Figure 1i and j). We used *Satb2* for staining upper-layer neurons (from layer II to layer IV; also expressed in callosally projecting neurons) and *Ctip2* for lower-layer neurons (layer V) as shown in previous studies⁷. Based on these images, cortical layering seems preserved in folded regions compared with controls, but layer thickness is reduced, particularly in the lower CP for sulcus areas (Fig. 5g).

6) In response to my previous minor point #5, the authors examined the number of bRG cells by counting the number of Pax6/PH3-double positive cells (new Figure 2f-j, Supplementary Fig. 2f, Supplementary Fig. 4e,f). However, Pax6/PH3-positive cells represented only dividing bRG cells rather than all bRG cells. The authors should count the number of Pax6-positive/Tbr2-negative cells in the SVZ to examine the number of bRG cells.

We agree that the number of basal Pax6/PH3-double positive cells does not reflect all bRG cells, and that it would be better to count the number of Pax6+/Tbr2+ cells in the SVZ. Unfortunately, staining both Pax6 and Tbr2 in the same section is challenging because the antibodies that work well in our hands for these proteins are from the same species (rabbit). Due to this technical limitation, we have rephrased the sentences in the results section to clarify that we examined the number of dividing bRG cells characterized by expression of PH3 and Pax6 by cells outside the VZ (Lines 180-182).

Minor comments

1) *Although the authors wrote as if Satb2 is an upper layer marker, Satb2 is expressed in callosally projecting neurons, which are also abundantly distributed in deep layers. The authors should modify their description of Satb2 throughout the manuscript.*

The Satb2 antibody is widely used for labelling upper cortical neurons (II-IV) during development⁸. Even though Satb2 also labels neurons with callosal projections, of which a fraction is present in layer V, most Satb2-positive cells reside in layer II to IV of the cortical plate during development⁸.

We have rephrased the first description of satb2 on page 6 to clarify this: Cortical layers were preserved as shown by Satb2 immunostainings, a marker for upper layer neurons (and callosally projecting neurons) and Ctip2, a marker for lower layer neurons. (lines 151-153)

2) *In response to my previous minor point #6, the authors added separate images (Supplementary Figure 5a). In the figure legend, although the authors wrote "Boundaries of upper and deeper layers are indicated", there were 3 kinds of lines, and it was unclear which was the boundary. The authors should describe what each of the line types represents.*

Agreed. We now indicate the location of the upper and lower cortical plate within the boundaries.

3) *As the antonym of "upper", the authors used both "deeper" and "lower" in the text. The authors should use "lower" and remove "deeper".*

Agreed. We have changed deeper to lower in main text and Figures.

Reviewer#2:

The authors have done a very nice job at addressing most of my comments on the previous version of this manuscript, and to my eyes their changes have improved the manuscript. However, and enthusiastic as I am about this study, I must insist in a couple of important issues in the manuscript that have not been properly addressed by the authors. I am sure that the authors will have no problem in addressing them, and thus my questioning must not be taken as a negative to have the manuscript published in Nat Comm; rather the opposite, I believe that the data of this important study must be presented with the best possible quality to later not be questioned once published.

1) *The authors have analyzed Pax6+PH3+ cells as suggested, but they counted also cells inside the VZ dividing non-apically. These cells are not bRG but subapical progenitors, as described in Pilz et al. 2013 (Nat Comm). While subapical progenitors are rare in the mouse cortex, since the authors did not distinguish them from real basal progenitors outside of VZ, it remains possible that these are the cells that increase in the triple mutants, instead of bRGs.*

We agree that the number of basal Pax6/PH3-double positive cells does not reflect all bRG cells within the SVZ. We have rephrased the results section to indicate that we estimated the number of dividing bRG cells characterized by expression of PH3 and Pax6 by cells outside the VZ. (lines 180-182)

2) Very importantly, the co-localization images are very far from convincing (actually some cells in several images of Gi 2 and Suppl Fig 2 are quite convincingly NOT double positive for Pax6/PH3, but indicated with arrowheads that they are) so I am not sure how were these could be analyzed and counted properly. The authors need to show some example detail images at higher magnification and proper z-resolution (i.e. single-plane confocal or apotome images).

Agreed. We now provide higher magnification of the images to better illustrate the Pax6/PH3-double positive cells with recounting to make sure we correctly counted the co-positive population (Figure 2f-j and Supplementary Figure 2g).

3) Again on the identification of bRG, I had previously indicate that bRG are recognized also by pVIM stain, but where pVIM labels any cortical mitotic cell (as shown by multiple authors previously) while bRGs are identified by the presence of a basal pVIM+ process. To support increase in bRG, the authors have now analyzed Pax6+/PH3+ cells outside of VZ. However, as mentioned in my second point now, the images they present are very far from convincing. Showing analysis of pVim+ cells with and without a basal process would definitely strengthen the central conclusion of this study that bRG abundance increases in their mutants.

We have quantified the number of basal pVIM cells with a basal process. Our results show that the percentage of sections with these cells is higher in *Cep83* tKO compared to control sections, where these cells are absolutely rare (see Figure 3 below). However, given that we do not include this data in the manuscript, and in line with the first point, we have removed the specification of bRG cells from the analysis of Pax6+/PH3+ cells outside of the VZ.

Figure. 3 : Quantification of pVIM+ cells with a basal process (marked with an arrowhead). Higher magnification images depict the basal process of pVIM+ cells in *Cep83* tKO embryos. Graph on the right shows total number of pVIM cells with a basal process in each genotype.

References

1. Shao W, *et al.* Centrosome anchoring regulates progenitor properties and cortical formation. *Nature* **580**, 106-112 (2020).
2. Sahara S, O'Leary DDM. Fgf10 regulates transition period of cortical stem cell differentiation to radial glia controlling generation of neurons and basal progenitors. *Neuron* **63**, 48-62 (2009).
3. Stahl R, *et al.* Trnp1 regulates expansion and folding of the mammalian cerebral cortex by control of radial glial fate. *Cell* **153**, 535-549 (2013).
4. Zhang X, *et al.* Clonal Analysis of Gliogenesis in the Cerebral Cortex Reveals Stochastic Expansion of Glia and Cell Autonomous Responses to Egfr Dosage. *Cells* **9**, (2020).
5. Del-Valle-Anton L, *et al.* Multiple parallel cell lineages in the developing mammalian cerebral cortex. *Sci Adv* **10**, eadn9998 (2024).
6. del Toro D, *et al.* Regulation of Cerebral Cortex Folding by Controlling Neuronal Migration via FLRT Adhesion Molecules. *Cell* **169**, 621-635.e616 (2017).
7. Chen B, Wang SS, Hattox AM, Rayburn H, Nelson SB, McConnell SK. The Fezf2-Ctip2 genetic pathway regulates the fate choice of subcortical projection neurons in the developing cerebral cortex. *Proc Natl Acad Sci U S A* **105**, 11382-11387 (2008).
8. Alcamo EA, *et al.* Satb2 regulates callosal projection neuron identity in the developing cerebral cortex. *Neuron* **57**, 364-377 (2008).

REVIEWER COMMENTS

Cortex Folding by Combined Progenitor Expansion and Adhesion-Controlled Neuronal Migration

Nature Communications manuscript NCOMMS-24-35775B

We would like to thank the reviewers for their careful reviews and comments which have helped us to further improve the manuscript. We have responded with a substantial amount of additional data and analysis, added more clarification where needed, and we revised text and figures accordingly. Changes in the main text are indicated in red.

Reviewer #1

The authors improved their manuscript significantly. However, they should address the following points before publication.

1) In my major point #2 in the first decision letter, I wrote as follows: "In Figures 2 and 4, the authors counted progenitor cells using dKO and tKO mice. The results seemed similar to those previously obtained using Cep83-KO and Fgf10-KO mice, and it was unclear what new information these results add compared to the previous results. It would be intriguing to examine the synergistic effects of Cep83-KO and Flrt1/3-KO on progenitor numbers, and those of Fgf10-KO and Flrt1/3-KO on progenitor numbers". However, the authors did not perform additional experiments to examine these synergistic effects, and therefore, it remained unclear how the information in Figures 2 and 4 adds to our conceptual understanding. The authors should explain this point, which does not necessarily have to be related to synergistic effects.

Figures 2 and 4 address the underlying mechanisms of the enhanced folding phenotype of *Cep83 tKO* and *Fgf10 tKO* mice. Previous work had indicated an increased density of Tbr2+ IPs in the subventricular zone of *Cep83 cKO* embryos and an expansion of apical progenitors in *Fgf10 cKO* embryos. Instead, *Flrt1/3 dKO* embryos had been shown to have normal numbers of progenitors.

In Figure 2, it was important to analyze, if the deletion of *Flrt1/3* in addition to *Cep83*, would alter the spectrum of progenitors in the developing cortex. We therefore performed quantifications in *Cep83 tKO* embryos of apical progenitors (Sox2+), IPs (Tbr2+), mitotic dividing cells (PH3+), Pax6/PH3-double positive cells, and Pvim+ cells with a basal process located outside the VZ. The results indicated that *Cep83 tKO* embryos had increased cell numbers of basal IPs (Tbr2+), Pax6/PH3-double positive cells. Moreover, we observed more Pvim+ cells with a basal process in *Cep83 tKO* embryos. Since the effect on IPs is already present in *Cep83 cKO* mice and is only mildly enhanced, if at all, by the deletion of *Flrt1* and *Flrt3*, it follows that the expansion of IPs and Pax6/PH3-double positive cells in *Cep83 tKO* embryos exacerbates the cortex folding phenotype seen in the *Flrt1/3 dKO* mice. We have changed the text of this paragraph to better explain the conceptual advance provided by Figure 2. (Lines 177-181)

In Figure 4, it was important to analyze, if the deletion of *Flrt1/3* in addition to *Fgf10*, would alter other types of progenitors besides APs. We performed quantifications of apical progenitors (Sox2),

IPs (Tbr2+), mitotic dividing cells (PH3), and Pax6/PH3-double positive cells in *Fgf10* tKO embryos. The results indicated that *Fgf10* tKO embryos had increased numbers of APs, but not IPs. These results suggest that the deletion of *Flrt1/3* in addition to *Fgf10*, did not alter other types of progenitors compared to the *Fgf10* single cKO mice. It follows that the expansion of APs in *Fgf10* tKO embryos exacerbates the cortex folding phenotype seen in the *Flrt1/3* dKO mice. We have changed the text of this paragraph to better explain the conceptual advance provided by Figure 4. (lines 238-253)

We have included the following sentences in the discussion to clarify this point (lines 395-400):

“Likewise, the increased numbers of IPs for Cep83 and APs for *Fgf10* reported in their single-KO lines^{18, 25} were similar to those observed in the tKO models, suggesting minimal effect by deleting *Flrt1/3*. Therefore, the combination of proliferation with divergent cell migration produces an enhanced cortex folding phenotype compared to the one produced by each of these two factors individually^{18, 25, 31}.”

2) As I wrote in my major point #3 in the previous decision letter, physiological folds are characterized by curvature at both the cortical surface and the border between gray and white matter, and curvature in both places appears at the same time during development. Although the authors wrote that previous papers used the term "folds" and "folding" even in the cases in which curvature appeared only at the cortical surface, it does not necessarily mean that the curvature in the previous papers really corresponded to genuine cortical folds. The authors should at least write that the phenotypes presented in this paper only partially recapitulate the features of physiological cortical folds, and it should be confirmed that the curvature truly corresponds to genuine cortical folds in future experiments.

We agree with the reviewer and now state in the results (lines 278-280): “These findings suggest that folding in *Fgf10* tKO and *Cep83* tKO mice partially recapitulate features of physiological cortical folds.” In addition, we state in the discussion (lines 448-449): “Additionally, the persistence of both sulci and gyri into postnatal stages provides the opportunities to confirm that the observed curvatures truly correspond to genuine and functional folds.”

Reviewer #2

Unfortunately, and for some unexplained reason, the authors continue to name subapical progenitors as basal progenitors. This mistake is very clear now that they nicely show detailed images: in Fig 2g the arrowhead points at a basal mitosis (border between Pax6+ VZ and Pax6-SVZ), but in Fig 2h the arrowhead points at a subapical mitosis (middle of the Pax6+ VZ). Hence, this is an incorrect account of basal mitoses (and potential basal bRG), and does not solve my first remaining issue. The authors, in spite of now stressing that they quantify “Pax6/PH3-double positive cells”, the section begins indicating that “We estimated the number of dividing bRG cells characterized by expression of PH3 and Pax6 by cells outside the VZ”. The naming and quantifications of Pax6+/PH3+ cells must be done and presented properly, or else the result is misleading.

We thank the reviewer for spotting this, and apologize for this mistake. The quantifications present in Figure 2f-j correspond to dividing cells at basal, not subapical, positions (Pax6/PH3-double positive cells). We have replaced the image in panel Figure 2h to indicate this.

Moreover, we have removed all references to bRG cells in the results section. Now, this section begins with the following sentence (lines 179-182): “We performed quantifications in *Cep83* tKO embryos of apical progenitors (Sox2+), IPs (Tbr2+), mitotic dividing cells (PH3+), Pax6/PH3-double positive cells, and Pvim+ cells with a basal process located outside the VZ³⁶ (Fig. 2; Supplementary Fig. 2).”

The authors nicely looked into pVim+ cells with a basal process, as I suggested previously, and identified such cells outside of VZ in some Cep83 tKO embryos, as shown in Fig 3 of the rebuttal letter. Given the above controversy about the identification of bRG with Pax6 and PH3 stains, these results should be included in the manuscript in that same section, to further support the increased presence of bRGs. Of note, the histogram shown in Fig 3 of the rebuttal letter seems somehow wrong, as in mutants there are individual datapoints at 0 and other at 1, so the average is not 1 as represented by the bars.

Thanks for the suggestion. We have updated the bar graph to indicate properly the average and we have added it to Supplementary Figure 2g. We now state in the results sections (line 183-187): “At both rostral and medial locations, we observed significantly increased cell numbers of basal IPs (Tbr2+) and Pax6/PH3-double positive cells compared to controls, consistent with previous findings in *Cep83* single KO¹⁸ (Fig. 2). We also observed more Pvim+ cells with a basal process in *Cep83* tKO embryos (Supplementary Fig. 2g).”